# Regulation of Membrane Calcium Transport Proteins by the Surrounding Lipid Environment

**DOI:** 10.3390/biom9100513

**Published:** 2019-09-20

**Authors:** Louise Conrard, Donatienne Tyteca

**Affiliations:** CELL Unit, de Duve Institute and Université Catholique de Louvain, UCL B1.75.05, Avenue Hippocrate, 75, B-1200 Brussels, Belgium; louise.conrard@uclouvain.be

**Keywords:** calcium exchanges, non-annular lipids, annular lipids, cholesterol, sphingolipids, acidic phospholipids, lipid domain, cell signaling, membrane curvature, membrane thickness, membrane lipid packing

## Abstract

Calcium ions (Ca^2+^) are major messengers in cell signaling, impacting nearly every aspect of cellular life. Those signals are generated within a wide spatial and temporal range through a large variety of Ca^2+^ channels, pumps, and exchangers. More and more evidences suggest that Ca^2+^ exchanges are regulated by their surrounding lipid environment. In this review, we point out the technical challenges that are currently being overcome and those that still need to be defeated to analyze the Ca^2+^ transport protein–lipid interactions. We then provide evidences for the modulation of Ca^2+^ transport proteins by lipids, including cholesterol, acidic phospholipids, sphingolipids, and their metabolites. We also integrate documented mechanisms involved in the regulation of Ca^2+^ transport proteins by the lipid environment. Those include: (i) Direct interaction inside the protein with non-annular lipids; (ii) close interaction with the first shell of annular lipids; (iii) regulation of membrane biophysical properties (e.g., membrane lipid packing, thickness, and curvature) directly around the protein through annular lipids; and (iv) gathering and downstream signaling of several proteins inside lipid domains. We finally discuss recent reports supporting the related alteration of Ca^2+^ and lipids in different pathophysiological events and the possibility to target lipids in Ca^2+^-related diseases.

## 1. Introduction

Membranes provide interfaces that not only separate two aqueous environments but also contribute to several functions, including regulation of solute exchanges, signal transduction, lipid metabolism, and membrane fusion and fission. To fulfill these roles, membranes must be tough and plastic at the same time. This could explain why membranes exhibit such a large variety of lipid species, and why they are arranged in far more intricate structures than simple homogenous fluid bilayers. Such membrane heterogeneity is illustrated by unequal lipid distribution at four different levels, that is, among (i) different cells, (ii) distinct intracellular compartments (e.g., endoplasmic reticulum (ER) vs. plasma membrane (PM)), (iii) inner vs. outer membrane leaflets (i.e., transversal asymmetry), and (iv) the same leaflet (i.e., lateral heterogeneity into lipid domains). Heterogeneity in local membrane lipid composition in turn generates areas of differential biophysical properties (e.g., lipid order, curvature, thickness) that could help to recruit/exclude and/or activate/inactivate specific membrane proteins, thereby participating in the spatiotemporal regulation of dynamic cellular events. 

In this review, we focus on calcium (Ca^2+^) transport proteins. Indeed, Ca^2+^ ions highly contribute to the cell physiology and biochemistry. They are one of the most widespread second messengers used in signal transduction pathways. They also act in neurotransmitter release from neurons, in contraction of all muscle cell types and in fertilization. Many enzymes require Ca^2+^ ion as a cofactor, including several coagulation factors [1]. Ca^2+^ ions are released from bone (the major mineral storage site) into the bloodstream under controlled conditions and are transported through bloodstream as dissolved ions or bound to proteins such as serum albumin. Substantial decrease in extracellular Ca^2+^ ion concentrations (hypocalcemia) can affect blood coagulation and even cause hypocalcemic tetany, characterized by spontaneous motor neuron discharge. On the other hand, hypercalcemia is associated with cardiac arrhythmias and decreased neuromuscular excitability. Moreover, upon excessive influx, Ca^2+^ ions can damage cells, possibly leading to cell apoptosis or necrosis. This is the case in excitotoxicity, an over-excitation of the neural circuit that can occur in neurodegenerative diseases, or after insults such as brain trauma or stroke [2]. Ca^2+^ ions also represent one of the primary regulators of osmotic stress.

Free Ca^2+^ cytoplasmic concentration is kept quite low at resting state (10–100 nM) in comparison to the ER/SR (endoplasmic/sarcoplasmic reticulum) (60–500 µM) [3,4] and the extracellular medium (1.8 mM) [5]. Ca^2+^ signals are generated within a wide spatial and temporal range through a large diversity of Ca^2+^ transport proteins, including channels at the PM upon response to extracellular stimuli as well as from the ER/SR or the mitochondria (not described in this review). The Ca^2+^ spike shortness in the cytoplasm is allowed thanks to the PM Na^+^/Ca^2+^ exchanger (NCX), the PM Ca^2+^ pump (PMCA), and the ER/SR Ca^2+^ ATPase (SERCA). 

Ca^2+^ transport proteins have been proposed to interact with, and to be possibly modulated through, the surrounding lipids. In general, those interactions can be classified according to the relative “residence” time of a particular lipid at the protein–lipid interface [6]. If a lipid displays a low degree of interaction with the protein transmembrane domain (TMD), it exhibits a fast exchange rate with lipids in close proximity and is considered as a “bulk” lipid (red in Figure 1A). Increased retention around the protein can result from specific interactions between the protein and the lipid polar headgroup, hydrophobic matching to the lipid hydrocarbon chains and creation of a membrane curvature, a.o. Such interactions reduce the exchange rates with the lipids and lead to the formation of a shell of “annular” lipids that surrounds the membrane protein (green in Figure 1A) [7]. For large, multiple transmembrane (TM)-spanning proteins, the composition of this shell can be heterogeneous, because the interactions depend on the local architecture of the membrane protein and its compatibility with the various lipids [8]. This immobilizing effect of the protein might extend beyond the first shell of directly interacting annular lipids (orange in Figure 1A), leading to further outer shells with a lesser extent of lipid immobilization [9]. Lipids with even lower exchange rates are denominated as “non-annular” lipids (blue in Figure 1A). These lipids are buried within the protein and generally directly interact with distinct hydrophobic sites of membrane proteins, named lipid pockets. Those lipids may fulfill diverse functions, from structural building blocks to allosteric effectors of enzymatic activity. 

Besides those three modes of interactions, a fourth still hypothetical way has emerged. It implies the gathering and downstream signaling of several proteins inside lipid domains (Figure 1B) which can be quite diverse. Among those are the lipid rafts, in which sphingolipids (SLs) form detergent-resistant membranes (DRMs) enriched in cholesterol and glycosylphosphatidylinositol (GPI)-anchored proteins in cold non-ionic detergents [10]. Recent progress in microscopy (such as combined fluorescence correlation spectroscopy (FCS) with stimulated emission depletion microscopy (STED) [11] or super-resolution microscopy [12]) provides evidence for the existence of such transient, nanoscale, cholesterol-, and SL-enriched membrane clusters. In addition to rafts, other nanoscale domains have been described at the PM of eukaryotes (i.e., caveolae and tetraspanin-rich domains [13,14]). Those lipid domains are controlled by lipid-based mechanisms through cohesive interactions. Lipid and protein clustering into lipid domains can also be regulated by protein-based mechanisms, involving specific interactions between lipids on one hand and membrane or cytoskeletal proteins on the other hand. Interaction between membrane and the cortical cytoskeleton is supported by experimental data from several groups and is integrated in the picket and fence Kusumi’s model (for additional information, see [15,16]). In agreement with the lipid-based independent mechanisms for lipid domain biogenesis, stable submicrometric lipid domains of different composition and biophysical properties than lipid rafts have been reported in artificial [17,18,19] and highly specialized biological membranes [18,20], and a variety of cells from prokaryotes to yeast and mammalian cells [21,22,23,24,25,26,27].

Thanks to cell imaging data, crystallography, and molecular simulations (Section 2), we summarize the recent advances indicating that Ca^2+^ transport proteins are modulated by their surrounding lipid environment (Section 3). We then provide an integrated view on the main modes of protein–lipid interactions (Section 4). Finally, we highlight recent reports supporting the related deregulation of Ca^2+^ and lipids in different pathophysiological events and the possibility to target lipids in Ca^2+^-related diseases (Section 5 and Section 6).

## 2. Overview of Models and Approaches Dedicated to Study Lipid–Protein Interactions

Even though the study of lipid–protein interactions has considerably progressed in the last decades, individual flaws of current techniques still need to be compensated by the overlay of several complementary approaches. 

### 2.1. Cell Imaging 

#### 2.1.1. Cell Imaging Approaches and Lipid Tools

The development of powerful imaging approaches is essential to evaluate the potential interaction between proteins and the surrounding lipids. Those approaches have to present a high resolution and to be compatible with live cell imaging for two reasons: (i) Lipid imaging should avoid cell fixation and/or permeabilization, as those processes could redistribute membrane lipids; and (ii) the study of membrane protein–lipid interaction is not relevant on fixed cells. 

Whereas laser scanning confocal microscopy (LSCM) does not provide sufficient resolution to explore protein–lipid interaction, the recent development of super-resolution microscopy techniques can help. Those include: (i) Photo- activation localization microscopy (PALM) and stochastic optical reconstruction microscopy (STORM), which use photoswitchable fluorescent probes to reveal spatial differences between molecules; and (ii) structured illumination microscopy (SIM), which relies on a grid pattern residing in one of the illumination apertures to generate a sinusoidal excitation wavefield that can be used to extract information from the image focal plane. However, these techniques are so far difficult to apply to living cells due to phototoxicity and photobleaching. Moreover, they are not able to simultaneously integrate high-resolution, sensitivity, and speed, precluding analysis of membrane protein–lipid interactions from a dynamical point of view. For advantages and limitations of super-resolution microscopy, see [16].

The Fast Airyscan imaging opens up new avenues to explore this question and appears very promising [28,29,30]. Airyscanning is a new detection concept that uses an array detector to oversample each Airy disk in order to gain sensitivity, resolution, and speed. In this way, Airyscanning achieves resolutions comparable to an extremely small pinhole, as in confocal microscopy, but with a much better signal-to-noise ratio. Such equipment allows to integrate sample analysis at both the cellular and molecular levels. It should also be more efficient than LSCM for a number of applications related to dynamic live cell imaging, such as ratio imaging, FRAP (Förster recovery after photobleaching) and FRET (fluorescence resonance energy transfer; see Section 2.1.2).

Another difficulty, besides the imaging approaches, is the limitation of reliable fluorescent tools for membrane lipid imaging. For a long time, the lipid field was limited to the insertion in the PM of fluorescent lipid probes obtained by grafting a fluorophore (such as nitrobenzoxadiazole (NBD) or boron-dipyrromethene (BODIPY)) on the lipid headgroup or its fatty acyl chain [16]. Although useful and easy, this technique could present limitations since the probes can differentially partition as compared to endogenous lipids. Polyene lipids (i.e., fluorescent lipids containing several conjugated double bonds) could circumvent the above difficulty since they have a structure highly similar to their endogenous counterparts and distribute within the cell together with their physiological kin [31,32]. Although suitable for in vitro and live cell analysis, those probes nevertheless also present some drawbacks, including a low quantum yield and high sensitivity to photobleaching [31]. During the last decades, innovative approaches dedicated to the analysis of endogenous lipid organization have been developed. They are based on fluorescent toxin fragments, proteins with phospholipid (PLP) binding domains, or nanobodies (for a review on these different classes of probes, see [16]).

#### 2.1.2. Förster Resonance Energy Transfer

FRET is highly suitable for the study of protein–lipid interaction. It is based on the energy transfer from an excited donor to an acceptor whose excitation spectra is covered by the emission spectra of the donor. The transfer is only possible when the two fluorescent molecules are extremely close, allowing to study molecular interactions. Again, lipid probes represent a limiting factor for this type of experiments. Polyene lipids presented in Section 2.1.1 have been used by Brugger et al. to analyze the interaction between the COPI (Coat Protein complex I) machinery TM protein p24 (in maltose-binding protein (MBP) fusion form) and SL species by FRET in a liposome-based assay. This study revealed a direct and highly specific interaction of exclusively one sphingomyelin (SM) species (i.e., SM with 18C) with the TMD of p24. Strikingly, the interaction depends on both the SL headgroup and the backbone and on a signature sequence within the TMD [33]. FRET using pyrene-labelled PLPs has also been used in micellar systems to determine the affinity between PLPs and the hydrophobic surface of the TM section of the PMCA, and to correlate those interactions with the thermal stability of the pump [34]. Alternatively, single-pair FRET can be combined with nanodiscs of specific composition (see Section 2.2), for example, to indirectly analyze the influence of specific lipid species on protein dimer formation. Thanks to this system, Sako et al. have revealed the importance of lipid (especially acidic lipid)–protein interplay in dimerization of juxtamembrane domains of EGFR [35].

A derivative from the FRET approach is the quenching of protein tryptophan fluorescence by halogenated (i.e., mainly bromylated) PLPs [36]. Those lipids behave much like conventional PLPs with unsaturated fatty acyl chains, but show significant absorption at the wavelength of tryptophan emission when localized directly around the residue. This technique has been used to show that SERCA1, which contains 13 tryptophan residues, has a higher binding constant for phosphatidylcholine (PC) than for phosphatidylethanolamine (PE) [37]. However, it is not possible to determine whether PEs bind less well than PCs at all sites or only at some specific sites. On the other hand, the group of Lee has introduced tryptophan residues in specific regions of prokaryotic mechanosensitive channels (MSC), which normally do not contain any tryptophan residues. Thanks to this localized insertion, they have determined the exact binding sites for several lipids at the inner and the outer PM leaflets [38].

### 2.2. Reconstitution of Membrane Proteins in Artificial Environments

Reconstitution of Ca^2+^ transport proteins in a controlled membrane environment represents a model of choice to study the importance of lipids for their activity. Proteoliposomes (i.e., spherical vesicles composed of a lipid bilayer and the protein of interest) are widely used. Although the composition of such liposomes has evolved to mimic, as close as possible, the cell membrane composition, several drawbacks remain. First, the membrane protein solubilization often results in turbid and viscous samples, which may be especially troublesome in many biophysical methods. Second, as liposomes are spherical, they present a membrane curvature that may interfere with the normal Ca^2+^ channel activity and this problem is especially relevant for curvature-sensitive channels like MSCs (see Section 4.3.3). This can be prevented by the use of planar bicelles rather than liposomes, which combine a flat bilayer-like plan and curved micelle-like ends. However, planar bicelles have a rather low structural integrity and stability.

Therefore, one should instead sometimes favor a stable planar bilayer model system of ~10 nm in diameter, which provides space for one or more membrane proteins and allows access to both sides of the bilayer for the association of substrates or signaling partners. As such, nanodiscs represent a robust tool for revealing the structure and function of isolated membrane proteins, as well as their complexes with other proteins and lipids [39]. Those self-assembled discoidal fragments of lipid bilayers can be controlled for their lipid composition and obtained in high yield. They are stabilized and rendered soluble in aqueous solutions by two amphipathic helical membrane scaffold proteins (MSP) (Figure 2) [40,41]. Their diameter (8–16 nm) is determined by the MSP length and the stoichiometry of the lipids used in the self-assembly process. The MSP encircles the discoidal membrane fragment in a “double belt” configuration, as initially suggested by [42]. Moreover, scaffold proteins are available with a wide range of specific tags/anchors for isolation, in vivo targeting, imaging, and reversible or irreversible surface immobilization [43].

The most direct method for incorporating integral membrane proteins into the nanodisc bilayer is to self-assemble it from a detergent-solubilized mixture of MSP and lipids, maintaining the correct overall stoichiometry of the component parts (Figure 2). Moreover, the size and ratio of scaffold protein-to-membrane protein may be selected to favor incorporation of predominantly monomeric (in which there is a large excess of scaffold protein and lipids) or oligomeric target proteins into nanodiscs [41]. Problematically, this necessitates the isolation and purification of the target protein, a process often leading to its inactivation due to separation from its native environment. To circumvent this problem, one could detergent-solubilize a native membrane preparation in the presence of excess MSP and lipids and then remove the detergent, which would distribute the starting membrane protein population into individual nanodiscs, thus forming a soluble library of all membrane proteins of the starting preparation [44]. This allows avoiding the purification step while accelerating the entire process of nanodisc self-assembly, which is sometimes critically important for preservation of the native and functional form of the target membrane protein. Proteins of many types, topologies, and sizes (from 1 to 24 TM domains) have been successfully self-assembled into nanodiscs [41]. 

Nanodiscs have already been used in combination with several techniques. For example, using nuclear magnetic resonance (NMR) or (cryo-)electron microscopy, one can evaluate the impact of lipid composition and biophysical properties on the protein orientation in the membrane, as well as its conformation changes. Using the latter technique, the transient receptor potential channel TRPV1 atomic structures of three conformations have been assessed, revealing locations of some annular and regulatory lipids that form specific interactions with the channel [45]. Besides conformation studies, nanodiscs are also very suitable to analyze, by spectroscopic methods, the activity of a single enzyme/channel embedded in a nanodisc of controlled composition [46,47]. A third application relies on the solubilization of an entire cell membrane in nanodiscs, followed by the immunoprecipitation of nanodisc-containing specific embedded lipids [48,49]. The embedded proteins that partition into those specific nanodiscs can then be identified (for example by mass spectrometry [50] or X-ray crystallography analysis). The reader could refer to Section 2.3 for a brief description of those methods.

### 2.3. Membrane Protein Structural Biology and Mass Spectrometry

Since the breakthrough in crystallizing membrane proteins [51], the most powerful method to define lipid-binding sites on proteins has been X-ray and electron crystallography [52]. This method is based on the fact that crystals of membrane proteins usually contain lipid molecules (see Figure 3). They originate from the native membrane, from which they co-purify with the crystallized membrane protein. Most of the lipids resolved in high-resolution crystal structures of membrane proteins are likely to be non-annular lipids (i.e., specifically interacting with the protein, see Section 4.1 and example in Figure 3a), their strong binding to the protein leading to immobilization of at least part of the lipids [6,53]. Other lipids, considered as annular lipids (Section 4.2 and Figure 3b,c), might also co-crystallize because they are needed to maintain the stability of the membrane protein. Indeed, lipids have been shown to play a major role for TM protein folding and stabilization as they mediate between the protein and the bulk lipids and seem to play a major role in the orientation of the membrane-spanning domain within the bilayer [54]. Upon analysis of crystallized proteins by X-ray or electron microscopy, these lipids appear in the density map as elongated structures mainly oriented perpendicular to the membrane plane. Specific binding sites on the protein can thus be identified and some lipid species can be characterized, even though they might not appear sufficiently definite to allow for their unambiguous characterization. Therefore, many lipid classes and molecular species can not be determined unequivocally. Progressively, more and more membrane protein X-ray structures show how lipids bind. For an extensive review describing the method development and applications, see [53]. Focusing on Ca^2+^ transporters, electron density maps for crystals of the SERCA in different states were obtained by X-ray solvent contrast modulation. They allowed to evidence PLP binding to the protein, either by Arg/Lys-phosphate salt bridges or by hydrogen bonding, and this binding appears to affect conformational switching [55].

Thus, X-ray or electron crystallography are increasingly used, not only to solve membrane protein structures with atomic resolution (till 0.5Å), but also to study their lipid environment [32]. However, those techniques still present huge limitations. The major one is that crystals of membrane proteins are mostly obtained from detergent solutions, which might alter the membrane protein stability and/or conformation. Moreover, they could also induce a biased sample of tightly bound lipids around the protein. This limitation may be overcome as more membrane protein structures are determined using crystals obtained from lipid cubic phases [56], allowing the crystallization of membrane proteins that never leave the lipid bilayer environment. Nanodiscs represent an extremely precious tool for this type of crystal formation [57,58].

In contrast to X-ray or electron crystallography, NMR spectroscopy is an analytical technique that bypasses the need for crystallization. Two types of NMR exist, namely solution and solid-state NMR. Solution NMR allows to study membrane protein–lipid interactions with high resolution, but only for small proteins that are embedded in small bicelles or nanodiscs. On the other hand, solid-state NMR has no theoretical limitations on size and, thanks to recent technical improvements like magic-angle spinning and cross polarization approaches, its resolution is now close to the solution NMR one [59,60]. Thus, solid-state NMR allows to prepare membrane proteins in lipid bilayers that more closely mimic their natural environment. However, practically, technical hurdles still limit the size of structures that can be solved by this method [61]. Several examples of protein–lipid interactions revealed by solid-state NMR are given in [62].

Cryo-electron microscopy relies on the instant freezing of a protein solution, leading to the formation of an amorphous solid without water molecule crystallization. The frozen sample is then screened and, thanks to the 2D images acquired, particle alignment and classification are carried out. The main advantages of this technique are that it requires only a small sample size and that proteins are maintained on a close-to-native state and are not required to be crystallized. However, it also presents defects like the quite low resolution of < 3.5 Å, that still renders it unusable for proteins with small molecular weight [63]. However, this resolution is increased in recent studies to up to 2.9 Å for TRPV1 and 2.2 Å for the β-galactosidase [64]. Cryo-electron microscopy may be the structural analysis technique where the advantages of nanodiscs are most effectively used, as the nanodisc can be analyzed as a soluble “single particle”, removing the need of protein solubilization. Moreover, nanodisc use removes the usual difficulty of unknown orientations of the particle analyzed [65].

Mass spectrometry (MS) of solubilized protein–lipid complexes is also useful. The most widely-used approach relies on the capability to dissociate the lipids from a membrane protein complex in the gas phase (i.e., successive delipidation), which, together with lipidomics analysis on purified membrane protein preparations, allows identification of key lipids [66]. Briefly, membrane proteins, either extracted from or reconstituted in detergent micelles or in other membrane mimetics (like nanodiscs), are introduced into the mass spectrometer via nano-electrospray ionization. This soft ionization process can preserve non-covalent interactions under the appropriate conditions, making it ideally suited for the study of protein complexes. For native MS of membrane proteins, bound lipids appear as adduct peaks on individual protein peaks. The characteristic mass differences between the protein and adduct peaks can reveal the presence of lipids. However, mass alone does not provide a complete lipid identity. Multistage ion activation enables selection of lipid-bound protein ions and dissociation of lipid species in tandem MS experiments [67]. Derivations of this technique also allow to evidence lipids that are necessary for membrane protein oligomerization [68].

### 2.4. Molecular Simulation

In silico approaches are more and more used to analyze the molecular mechanisms of protein–lipid interactions [69,70]. The atomistic simulations of molecular mechanics used to be limited by the calculation times on the available computer systems. This has been partially overcome by the use of parallelized super computers and a variety of optimized simulation softwares, which enable simulations in atomistic details for systems. The simulation of molecular dynamics (MD) allows the analysis of the interactions of atoms and macromolecules for a short time period by the established laws of physics and can be considered as an animation of Newtonian mechanics. Unlike NMR- and X-ray-based approaches, the motions and interactions of proteins and lipids can be monitored in atomistic detail with high temporal and spatial resolution [32].

However, the computational cost of the simulations is such that length scales beyond a few microseconds are not currently readily accessible [71], especially for extended systems containing multiple membrane proteins. This is why we saw the emergence of more approximate coarse-grained (CG) representations of membrane lipids and proteins in MD simulations [72], in which groups of atoms are represented as single particles (Figure 4A). CG simulations can enhance lipid exploration of the protein surface and candidate binding sites, while sacrificing the finer detail of lipid–protein interactions. These approximations may be reconciled to some degree by conversion of the endpoint of a CG system back to atomistic detail [73,74] and subsequently running an atomistic simulation to assess the validity of the CG system arrangement, a so called (serial) multiscale modelling approach [75].

The structure of a membrane protein used as initial input for MD simulations may originate from X-ray or cryo-electron crystallography, or NMR. If the 3D structure of the protein is not known experimentally, a model may be built by modelling in some cases. The membrane protein is then embedded into a lipid bilayer [73]. This may be achieved either by self-assembly simulations [78], in which short simulations are run to allow the spontaneous formation of a bilayer around an integral membrane protein, or by a number of methods which insert a membrane protein into a pre-assembled bilayer [79,80]. Advances in lipid parameterization [81], along with a growing appreciation of the in vivo compositional and spatiotemporal complexity of lipid membranes, enable simulations of proteins in complex lipid bilayers, providing approximations of in vivo PM composition [82,83] and transversal and lateral asymmetry [84]. Such mixed lipid systems allow to address competition between different lipid species for interaction with a given protein, and provide a better approximation of lipid–lipid interactions which are linked to, and may influence protein–lipid interactions. Such mixed lipid systems can now be routinely assembled in CG.

A number of recent simulation studies probing lipid interactions have identified annular solvation shells around proteins [80] as well as specific lipid binding. These sites show good agreement with those identified from a range of structural studies. A number of other, presumably weaker, binding sites can also be resolved. While these weaker sites may not always be observed by X-ray crystallography, they can be probed through fluorescence, NMR, or MS. Overall, MD simulations have strong predictive power, are well-suited for identification of these sites, and could even characterize the identified sites through estimation of lipid-binding affinities while giving insight into mechanisms of lipid modulation. For example, MD simulations have allowed to show that TM pockets of the prokaryotic mechanosensitive channels of small conductance (MSCS) have a decreased volume upon channel opening, and that lysoPLPs (i.e., PLPs with one acyl chain per headgroup) displace normal PLPs from the pockets and trigger the channel opening (Figure 4B) [77,85].

## 3. Overview of Membrane Ca^2+^ Transport Proteins and Evidences for Their Modulation by Lipids

The majority of Ca^2+^ transport proteins has been proposed to be regulated by membrane lipids (Figure 5). As mentioned in the Introduction section, surface and intracellular membranes are quite complex in lipid composition as well as in lateral and transversal organization. Moreover, this complexity differs from mammalian cells as compared to yeast and bacteria. In mammalian cell membranes, one can distinguish three main lipid classes: the PLPs, SLs, and cholesterol. PLPs are divided in acidic and neutral ones, according to their polar head properties. Acidic PLPs include phosphatidic acid (PA), phosphatidylserine (PS), and phosphatidylinositol (PI), alone or conjugated to one (PIP) or two (PIP_2_) phosphates, a.o. Neutral PLPs consist of PC and PE. SLs are derived from ceramide (Cer), which is decorated with a phosphocholine headgroup in the case of SM or with saccharides in the case of glycoSLs. The latter include cerebrosides, gangliosides and globosides, a.o. In the yeast *S. Cerevisiae*, ergosterol is the predominant sterol and three types of SLs that are different from the mammalian ones can be found. They are built on a Cer backbone linked to an inositol phosphate headgroup, forming inositol phosphate Cer (IPC). The latter might be linked to a complex oligosaccharide, forming mannosylated IPC derivatives (MIPC). In contrast, the membrane of most bacteria is devoid of sterols but enriched in PE, phosphatidylglycerol (PG), and cardiolipin (CL). For a comparison between the PM compositions of different cells, see [16].

Besides differences in composition, membranes are susceptible to lipid enzymatic modifications, generating additional potential modulators for Ca^2+^ transport proteins. Phospholipases (PLs) hydrolyze PLPs. Four major classes are distinguished based on the type of reaction they catalyze. PLAs and PLB cleave acyl chains, either the first one (PLA_1_), the second one (PLA_2_), or both (PLB). PLA_2_ can produce arachidonic acid (AA) as well as lysoPLPs. On its side, PLC hydrolyzes the ester bond between the glycerol and the phosphate group, releasing a diacylglycerol (DAG) and a phospho headgroup, typically inositol-3-phosphate (IP_3_). Both DAG and IP_3_ released by PLC are important second messengers that control diverse cellular processes. While IP_3_ diffuses through the cytosol to bind and open IP_3_R channels at the ER membrane, DAG remains bound to the membrane and activates the protein kinase C (PKC). Finally, PLD cleaves after the phosphate, releasing PA and a headgroup [86]. Another enzymatic modification is achieved by sphingomyelinase, generating Cer, which can be subsequently hydrolyzed into sphingosine thanks to ceramidase, and both Cer and sphingosine can be phosphorylated by specific kinases. 

### 3.1. Ca^2+^ Influx in the Cytosol through Cation Channels 

The increase of the cytosolic Ca^2+^ concentration can result from the activation of cation channels at the PM, like transient receptor potential channels (Section 3.1.1), voltage-activated ion channels (Section 3.1.2), or mechanosensitive channels (Section 3.1.3), which respond to extracellular stimuli. The Ca^2+^ release into the cytosol can also originate from the ER/SR, through the ER/SR-associated inositol-3-phosphate receptor (IP_3_R) and ryanodine-receptor (RyR) channels (Section 3.1.4), and the PM-associated Orai1 channel (Section 3.1.5). 

#### 3.1.1. Transient Receptor Potential Channels

##### Overview and Pathophysiological Implications

Transient receptor potential (TRP) channels are a group of non-selective cation channels activated by a large panel of stimuli and that are expressed mostly at the PM of numerous cell types. The TRP family comprises more than 30 channels, most of which are permeable for Ca^2+^, but also for Mg^2+^ or Na^+^ [87]. They share some structural similarities to each other, most of them being composed of six membrane-spanning helices with intracellular N- and C-termini [88]. Based on sequence homology, the TRP family has been divided in seven main subfamilies: TRPC (C for “canonical”), TRPV (V for “vanilloid”), TRPM (M for “melastatin”), TRPA (A for “ankyrin”), TRPN (N for “no mechanoreceptor potential C”, not expressed in mammals), TRPP (P for “polycystin”), and TRPML (ML for “mucolipin”).

Although all TRP channels are molecular sensors, each sub-family is unique. Indeed, by acting as sensors of osmotic pressure, volume, stretch, and vibration, those channels mediate a wide range of sensory functions such as pain, temperature, different kinds of tastes, pressure, or vision. Moreover, even though they are not defined as voltage-gated (i.e., they do not require a change in membrane potential to open), they are often considered as voltage-sensitive (i.e., their activity is modulated by voltage). Additionally, they are emerging as polymodal ion channels, being sensitive to a multiplicity of mechanisms, activators, and inhibitors, suggesting that they may serve as integrative sensors of complex chemical signals [89,90]. The most extensively-studied TRP channel is TRPV1, which responds to stimulation by temperature, acidic conditions, capsaicin, and the pungent component of wasabi. It induces a painful, burning sensation and is mainly found in nociceptive neurons as well as the central nervous system, where it mediates the transmission and modulation of pain [91]. Mutations in TRP channels have been linked to several neurodegenerative disorders and skeletal dysplasia, but also kidney disorders, as extensively reviewed in [92]. Their altered expression also often leads to tumorigenesis [93,94]. On the other side, they represent therapeutic analgesic targets for reduction of chronic pain, as their antagonists reduce the sensitivity to stimuli of channels involved in nociception, like TRPV1 [95,96].

##### Regulation by Lipids

TRP channels have been shown to be modulated by acidic PLPs (e.g., PI(4,5)P_2_) and PLP derivatives (like AA, lysoPLPs, and DAG). For instance, PI(4,5)P_2_ appears to activate most TRP channels [97]. However, some of them (like TRPC4a, TRPV3) were shown to be inhibited [98,99] or to expose opposite behaviors depending on the experimental settings (like TRPC5), suggesting a complex regulation [100]. The effect of PI(4,5)P_2_ on TRPV1 remains largely debated and is thought to be dual [101,102]. As a matter of fact, as least two sites of regulation have been evidenced, one on the C-terminal segment [103], and the other in the capsaicin-binding pocket. The latter thussuggests a non-annular type of interaction, in which the PI acts as a competitive agonist and negative allosteric modulator, but may also function as positive, obligatory co-factor [45]. Concerning TRPM3, it also binds PI(4,5)P_2_ but no regulatory effect has been reported yet [104]. Altogether, those observations suggest that there are divergent and complex effects of PI(4,5)P_2_ on TRP channels. As suggested for TRPV1, but also TRPC6 and TRPC4a, effects may occur through direct binding at a site in the C-terminus where it might compete with calmodulin [105]. Finally, the effect on some TRP channels (like TRPC4) might also depend on the actin cytoskeleton and PDZ-binding motifs [98].

The PLP derivative AA and its many active metabolites are important stimulators of TRPC channels, most notably TRPC6 [106], but also of other TRP channels such as TRPA1 [107] and TRPM2 [108]. On the contrary, AA inhibits TRPM8 [109] and has no effect on TRPC5 [110]. Endovanilloids are derived from AA and are endogenous agonists of the pro-nociceptive TRPV1 channels [111]. 

On the other side, lysoPLPs, such as lysoPC (LPC), lysoPI (LPI), and lysoPA (LPA), also modulate TRP channels [112]. LPC and LPI have been shown to activate TRPC5, but not TRPM2, in HEK293 and smooth vascular cells. The regulation has been shown to not involve G-protein receptors or other co-factors, and has been hypothesized to be linked either to a direct binding or to modifications of the bilayer properties [110]. Another proposed way of activation of TRPC5 by LPC in endothelial cells involves TRPC6. Indeed, the latter’s activation by LPC would lead to the TRPC5 trafficking at the PM and activation, inducing a prolonged increase in intracellular Ca^2+^ concentration and subsequent decreased cell migration and healing in atherosclerotic arteries [113]. LPC and LPI also activate TRPV2, but this time through a G-protein signaling that induces translocation of TRPV2 at the PM. This activation has been linked to increased prostate cancer cell migration [114]. Finally, LPA, which accumulates under tissue damage and ischemic conditions and is involved in the generation of chronic neuropathic pain, can activate TRPV1 through a binding site in its C-terminal domain, indicating its contribution to pain transduction [115]. TRPM7 is, on the other side, inactivated by LPA [116]. All those results suggest that lysoPLPs may play important roles in a wide range of biological phenomena. 

Finally, several TRPC channels, including TRPC2/3/6/7, are activated by DAG and its analogs (e.g., 1-oleoyl-2-acetyl-sn-glycerol (OAG) or 1,2-dioctanoyl-sn-glycerol (DOG)), either exogenously added [117] or generated in response to PLC-coupled receptors [118]. Studies on TRPC6 suggest a direct action of DAG on these channels, rather than an indirect effect through the activation of PKC [119]. Nevertheless, a synergism with IP_3_ appears to occur, potentially conferring greater sensitivity to DAG [120]. In contrast, TRPC1 does not appear to be directly activated by DAG but could be phosphorylated by PKC [121], and TRPC4/5 are not activated by DAG [119,122]. TRPV1 and TRPA1 are also activated by DAG with different potencies and probably through direct binding. Indeed, a TRPV1 mutant channel unable to bind capsaicin does not react with DAG [123]. However, the concentrations required to produce these effects are quite high, suggesting the alternative possibility that DAG does not activate but, rather, “primes” both TRPV1 and TRPA1 for activation [124].

Aside from PLPs and derivatives, some SLs and derivatives also modulate TRP channel activity. Indeed, sphingosine, but not Cer, has been shown to activate TRPM3 on HEK cells analyzed by whole-cell patch clamp [125]. Those TRPM3 channels have been identified on oligodendrocytes, where sphingosine-mediated Ca^2+^ increase contributes to cell differentiation [126]. On the other side, sphingomyelinase treatment inhibits Ca^2+^ response through TRPV1, while neither Cer nor sphingosine influence the channel activation [127]. This suggests a modulation by reduction of SM content, rather than by the production of derivatives. Finally, sphingosine-1-phosphate has been shown to stimulate TRPC5 in vascular smooth muscle cells via two mechanisms, one extracellular and one intracellular, consistent with its bipolar signaling functions [128].

Cholesterol also appears to regulate TRP channels. This is based on the observation that depletion of membrane cholesterol by treatment with methyl-β-cyclodextrin (mβCD) differentially affects TRP channels, stimulating TRPM8 and TRPM3, while reducing the activity of TRPV4 and TRPV1 [118,127].

#### 3.1.2. Voltage-Gated Channels

##### Overview and Pathophysiological Implications

Voltage-gated ion channels (VGCs) are particularly important in excitable cells such as muscular cells and neurons. They are activated by changes in the electrical PM potential near the channel proteins, which alter their conformation, resulting in their opening and closing. Those channels are usually ion-specific and channels specific to Na^+^, K^+^, Ca^2+^, and Cl^−^ ions have been identified [129]. The biochemically-characterized voltage-gated Ca^2+^ channels (VGCC or Ca_v_ channels) are complex proteins made of four or five distinct subunits, which are encoded by multiple genes. The α_1_ subunit (Ca_v_α_1_) of 190–250 kDa is the largest subunit, containing 24 TM domains. It incorporates the conduction pore, the voltage sensor and gating apparatus, and the known sites of channel regulation by second messengers, drugs, and toxins. Therefore, this subunit determines the physiological and pharmacological properties of the Ca^2+^ current and, as a consequence, the nomenclature. 

Three major types of currents that correspond to three α_1_ subunit gene families have so far been described. First, L-type Ca^2+^ currents, which occur through Ca_v_1 channels, require a strong depolarization for activation and are long lasting. They are the main Ca^2+^ currents recorded in muscle and endocrine cells, where they initiate contraction and secretion. Second, N-, P/Q-, and R-type Ca^2+^ currents, which occur through Ca_v_2 channels, also require strong depolarization for activation. They are expressed primarily in neurons, where they initiate neurotransmission at most fast synapses. Third, T-type Ca^2+^ currents, which take place through Ca_v_3 channels, are activated by weak depolarizations and are transient [130]. As Ca_v_2-type channels, they are critically important for regulating neuronal excitability, both in the central and peripheral nervous system, and are essential mediators of hormone secretion [131]. Interestingly, some of those channels also appear to be mechanosensitive. For instance, Langton et al. reported that whole-cell inflation (as a membrane stress) induces smooth muscle L-type Ca^2+^ currents [132]. Since then, the mechanosensitivity of the L-type Ca_v_ channels has been tested in myocytes in more native conditions using several mechanostimuli-like fluid pressure [133] or axial stretching [134]. Thus, Ca_v_ channels regulate a wide variety of processes in excitable cells, such as contraction, secretion, and neurotransmission. Consequently, Ca_v_ channel dysfunction has been associated with several diseases. Notably, channel hyperactivity has been shown in several neurological disorders like chronic pain conditions and epilepsy, but also muscular and vision syndromes [135]. Therefore, several specific blockers are currently used as therapeutic agents [136]. On the other side, as Ca_v_ channels also express a mechanosensitivity, their activation in hypertension, which exerts a mechanical pressure on the cardiomyocytes, could elicit a Ca^2+^ signaling that leads to cardiac hypertrophy [137].

##### Regulation by Lipids

PI(4,5)P_2_ has been proposed to activate Ca_v_ channels in a voltage-independent pathway but the extent of this activation is not clear. In a patch clamp of excised patches containing P/Q- and N-type channels (i.e., Ca_v_2 channels), the current is completely and rapidly abolished by PI(4,5)P_2_ depletion [138]. On the other hand, in a whole-cell patch clamp of HEK293 cells expressing L-type (i.e., Ca_v_1) or P/Q-type channels, the current is only attenuated (~ 29% to ~ 55% of control current) upon PI(4,5)P_2_ depletion [139]. Ca_v_ current amplitude (mostly originating from L-type Ca_v_ channels) in pancreatic β-cells is also reduced upon PI(4,5)P_2_ depletion. This depletion is achieved through muscarinic M1-receptor activation or activation of a voltage-sensitive phosphatase [140,141]. Therefore, PI(4,5)P_2_ is expected to have a stimulatory effect on Ca_v_ channels, stabilizing their activity by reducing current rundown. Even if the molecular mechanism of regulation remains unclear, recent evidences reported that the subcellular localization of Ca_v_β subunit, and especially of its flexible “HOOK” region, is a key factor for the control of PI(4,5)P_2_ sensitivity via dynamic electrostatic and hydrophobic interactions with the PM [142]. In contrast to this activation role, an antagonistic effect of PI(4,5)P_2_, occurring at the same time, has also been suggested in some studies. For instance, PI(4,5)P_2_ produces a voltage-dependent inhibition by shifting the activation curve to more positive voltages [143]. This inhibition is thought to be alleviated by phosphorylation by cAMP-dependent protein kinase A (PKA) [138]. 

The PLP derivative AA also modulates Ca_v_ channel currents, but at more than one site, resulting in biphasic effects. Indeed, AA inhibits currents in Ca_v_1-3 on one hand, and acts at a second distant site to enhance N-current at negative potentials on the other hand [144,145]. On the other side, LPA (but not PA) has also been shown to induce a Ca^2+^ influx in red blood cells (RBCs). The channel involved has not been identified, but the entry can be prevented by P-type channel blockers, suggesting a role of the erythrocyte Ca_v_2.1 channels [146]. This activation induces the erythrocyte aggregation and could be involved in the thrombus formation, around which LPA concentrations are increased [147].

Regarding SL-mediated regulation, sphingomyelinase activity stimulates Ca_v_ channels expressed in oocytes [148], an effect that could be linked to the increased channel activity by Cer-1-phosphate in rat pituitary cells [149]. 

Finally and as exposed below (see Section 4.3.1), membrane fluidity and consequently cholesterol content also contribute to regulate Ca_v_ channel activity. 

#### 3.1.3. Mechanosensitive Ion Channels

##### Overview and Pathophysiological Implications

MSCs are located at the PM of many cells and are gated directly by physical stimuli. They transduce those stimuli into an electrical or chemical signal in hearing, touch, and other mechanical senses. MSCs vary in selectivity, from cation-selective to K^+^-highly selective in eukaryotes. In comparison to other receptors, the identification of mechanically-activated ion channels has been delayed due to several difficulties: (i) Mechanosensory cells are not abundant and usually mixed with other cells; (ii) MSCs are usually expressed at low levels; and (iii) the channels may be associated in complexes with auxiliary proteins [150].

The first well-studied MSCs were the prokaryotic MSCL (L for “large conductance”), MSCS (S for “smaller conductance”), MSCK (K for “K^+^-dependent smaller conductance”), and MSCM (M for “mini-conductance”). They contribute to protection of bacteria from hypo-osmotic shock [151], but also play roles during the cell wall re-modelling upon entry in, and exit from, stationary phase [152]. In eukaryotic cells, five different types of channels have been shown to transduce mechanical signals so far: (i) the degenerin/epithelial Na^+^ channels (DEG/ENaC), (ii) two-pore domain K^+^ (K_2P_) channels like TREK and TRAAK, (iii) MSCS-like channels, (iv) TRP channels, and (v) Piezo channels. The three last ones are non-selective for cations [150,153]. 

Since their discovery [153], Piezo channels have encountered a great interest among scientists. This family of proteins comprises two members in human, Piezo1 and 2. The genes coding for those proteins share 47% identity and have no similarity to any other protein. Piezo channels are trimers, each mouse protomer containing at least 26 TM helices, that are shaped like a propeller with three curved blades organized around a central pore. The flexible propeller blades can adopt distinct conformations, and consist of a series of four TM helical bundles (i.e., the Piezo repeats) [154]. Piezo1 is a sensor of mechanical forces in endothelial, urothelial, and renal epithelial cells. It is also involved in blood vessel development and integrity, as well as in RBC volume homeostasis. Piezo2, on the other hand, is majorly expressed in somatosensory neurons, where it regulates proprioception and lung inflation-induced apnea [155]. As for other Ca^2+^ channels, stretch-activated channels have been associated with major pathologies including cardiovascular diseases, Duchenne muscular dystrophy, lymphatic dysplasia, distal arthrogryposis, and RBC diseases like stomatocytosis [156,157,158,159]. Moreover, as their discovery is still recent, the extent to which they regulate development and physiology is yet to be completed.

##### Regulation by Lipids

MSCs are known to be modulated through PI(4,5)P_2_ and PI(4)P. As a matter of fact, depletion of both species leads to the inactivation of Piezo1 and 2 [160]. This depletion can be induced experimentally either by PI phosphatases or by the activation of TRPV1 channels through capsaicin. Such treatment indeed induces a Ca^2+^ increase, which in turn activates the PLC isoform δ, leading to the degradation of both PI(4)P and PI(4,5)P_2_ [161]. This capacity of TRPV1 activation to induce Piezo channel inhibition could partly explain the analgesic effects of capsaicin. On the other hand, activation of the PLC isoform β (through G-protein coupled receptors), an isoform which only targets PI(4,5)P_2_ but not PI(4)P, only marginally inactivates Piezo currents [160]. Acidic PLPs also appear important for the regulation of the prokaryotic MSCs [162] and of the mechano-gated K^+^ channel TREK-1 [163]. Moreover, PS exposure at the surface of myoblasts suppresses piezo1 activation and in turn impairs myotube formation, suggesting that the appropriate localization of PS in the inner PM leaflet is crucial for the activity of piezo1 and that cell surface flip-flop of PS acts as a molecular switch for piezo1 activation that governs proper morphogenesis during myotube formation [164].

Regarding the PLP derivatives, lysoPS (LPS) suppresses Ca^2+^ influx via Piezo1 when it is incorporated into the outer PM leaflet of C2C12 myoblasts. On the other side, incorporation of LPC (a zwitterionic lysoPLP) shows no significant inhibitory effect on Ca^2+^ influx. All those data suggest that the PLPs with a phosphoserine headgroup present on the outer PM leaflet are responsible for the inhibition of piezo1 activation [164].

Finally, the activity of MSCs also appears to be regulated by lipid rafts and thus by cholesterol (see Section 4.4).

#### 3.1.4. Endo/Sarcoplasmic IP_3_R and RyR Channels

##### Overview and Pathophysiological Implications

The release of Ca^2+^ ions from the ER/SR is the most versatile cellular signaling mechanism and is responsible for several cellular effects including secretion [165], smooth muscle contraction [166], gene transcription [167], and fertilization [168]. Ca^2+^ release from those stores is mediated by two Ca^2+^-release channels (i.e., the inositol-1,4,5-triphosphate receptors (IP_3_Rs) and the ryanodine receptors (RyRs)), the latter being so named because they possess a high-affinity binding site for the plant alkaloid ryanodine [169]. The IP_3_R family is ubiquitously expressed and is comprised of three homologous isoforms (types 1–3) encoded by separate genes. The IP_3_R is formed of four 313 kDa-subunits that comprise three major regions: The N-terminal region to which IP_3_ binds, the C-terminal region with its six TM domains, and a large intervening sequence [170]. RyR channels have similarities with IP_3_Rs. First, the RyR family is also comprised of three isoforms, type 1 being expressed in skeletal muscle, type 2 in both cardiac muscle and brain, and type 3 at low levels in a wide range of tissues [171]. Second, the RyR complex is also a tetramer, although a little bigger than the IP_3_R complex. Third, the pore of both channels is formed by the final pair of TM domains (domains 5–6) and the luminal loop that links them from each of the four subunits [172,173].

IP_3_R and RyR channels are extremely efficient Ca^2+^-cytosolic release channels with a large conductance, but only modest selectivity for Ca^2+^ over monovalent cations [173,174]. In the ER/SR, where most channels are located, this lack of selectivity is probably not problematic since Ca^2+^ exhibits an appreciable electrochemical gradient across the ER membrane. As their activation follows an increase of the Ca^2+^ concentration in the cytosol, IP_3_R and RyR most probably do not contribute to the general enhancement of the Ca^2+^ concentration, but are rather responsible for complex patterns relative to both space and time, such as Ca^2+^ waves, oscillations or sparks [175]. For instance, “nuclear Ca^2+^ waves” that engulf the entire nucleus without spreading into bulk cytosol have been evidenced upon IP_3_R/RyR activation in cardiomyocytes [176], and could activate Ca^2+^-dependent transcription factors [177]. Ca^2+^ sparks could also contribute to membrane hyperpolarization through the activation of plasmalemmal BK_Ca_ (large conductance Ca^2+^-activated K^+^ channels) [178]. Dysfunction of these ion channels has been linked to a wide range of neurodegenerative and neuromuscular diseases like ataxia, but also heart disease, exocrine secretion deficit, and taste-perception deficit [179,180].

##### Regulation by Lipids

Besides Ca^2+^ [174], phosphorylation [181], ATP [182], and numerous interacting proteins [183], IP_3_R channel activity is regulated by IP_3_, which is produced upon PLC activation by a variety of G protein-coupled receptors (GPCRs), including muscarinic, adrenergic, and angiotensin receptors [184]. PI(4,5)P_2_ also regulates the ER/SR Ca^2+^ transport proteins by stimulating RyR while inhibiting IP_3_R activity. This has been mostly tested by incorporation of the receptors into membrane bilayers containing, or not containing, PI(4,5)P_2_ [185,186]. However, the physiological relevance of this regulation can be asked, as PI(4,5)P_2_ concentration at the ER/SR membrane is even lower than at the PM.

#### 3.1.5. Store-Operated Ca^2+^ Release-Activated Ca^2+^ (CRAC) Currents

##### Overview and Pathophysiological Implications

At rest, the SERCA activity is sufficient to maintain high [Ca^2+^]_ER_, but after elevated Ca^2+^ signaling and release of ER-associated Ca^2+^ into the cytoplasm, the ER store may become depleted. The emptying of the ER stores is sensed by stromal interaction molecules (STIM) [187] that interact with Orai1 proteins (also called CRACM1) at the PM [188,189], allowing for SOCE (for store-operated Ca^2+^ entry) currents mediated by Orai1 channels (i.e., CRAC [for store-operated Ca^2+^ release-activated Ca^2+^] currents).

The STIM protein family includes two members, STIM1 and STIM2, which share ~ 61% sequence identity [190]. Those single TM proteins contain (i) a N-terminal portion including the Ca^2+^-sensing domain localized within the ER lumen, and (ii) a long cytosolic strand, which couples to Orai channels in the PM. The Ca^2+^-sensing domain is formed by an EF-hand (i.e., a low affinity Ca^2+^ binding domain which loses Ca^2+^ upon store depletion). The transmission of this loss is ensured through the sterile-α-motif (SAM) domain, which initiates the CRAC signaling cascade [191]. Thus, the Ca^2+^ depletion leads to a change of conformation of STIM protein in its active form [192]. STIM1, normally homogenously distributed [193], moves rapidly to multimerize while translocating along the ER membrane towards a location where the ER and the PM are in close proximity (i.e., a ~ 17 nm distance) [194]. Subsequently, STIM1 couples to and stimulates Orai1, initiating pronounced CRAC currents [195]. However, STIM1 function is likely to be even more complex, since it has been shown to not only activate Orai1 but to also inhibit neighboring voltage-activated Ca_v_1.2 channels [196] and to activate TRPC1 channels [189].

Orai proteins are highly selective Ca^2+^ channels present at the PM [197]. The three homologs, Orai1–3, contain four TM domains that share ~ 81%–87% sequence identity, flanked by cytosolic N- and C-termini. Both cytosolic strands are required for functional coupling to STIM1 [198]. Thanks to its crystal structure, unveiled in 2012, the channel has been shown to be composed of six Orai subunits, with the TM domains arranged in concentric rings around a central aqueous pore formed exclusively by the first TM helix of each subunit. Moreover, Orai monomers are arranged as three dimers, leading to a three-fold symmetry [199]. Mutations in the STIM–Orai proteins have been linked with severe combined immune deficiency syndrome [188]. Indeed, Orai channels play an important role in the activation of T-lymphocytes.

##### Regulation by Lipids

PI(4,5)P_2_ appears critical for the SOCE mechanism and especially for the STIM1 interaction with the PM. Indeed, PI(4,5)P_2_ depletion partially inhibits translocation of STIM1 into puncta, a process that can be recovered by overexpression of Orai1 [200]. Accordingly, upon activation, the exposed C-terminal polybasic segment of STIM1 interacts with PI(4,5)P_2_ or PI(3,4,5)P_3_ at the PM, trapping STIM1, which in turn binds to Orai1 [201]. Additionally, the Ca^2+^-independent iPLA_2_β has been shown to be an essential component of signal transduction for the SOCE mechanism in several cells [202]. As a matter of fact, lysoPLPs products of iPLA_2_β activate SOCE in excised membrane patches, suggesting a STIM-independent pathway [203].

SLs and derivatives also regulate the SOCE mechanism. Indeed, Orai1 activity is highly decreased by a sphingomyelinase treatment, suggesting a positive effect of SM [204]. On the other side, Cer induces activation of CRAC channels [205], an observation that apparently contrasts with the study showing that the addition of Cer-1-phosphate does not modulate the Orai1 activity [204].

Finally, Orai1 and its modulator STIM are also thought to be directly regulated by cholesterol [206,207]. In fact, cholesterol has been shown by fluorescence binding studies to bind to the Orai1 amino-terminal fragment. The same group has revealed that cholesterol binds to the full-length Orai1 in mucosal-type mast cells, allowing them to conclude that Orai1 might sense the amount of cholesterol in the PM and their interaction might inhibit the channel activity, thereby limiting SOCE. As a consequence, cholesterol depletion enhances Orai1-mediated CRAC currents in HEK cells expressing Orai1 as well as in mast cells [207]. Another study shows contradictory results on pulmonary endothelial cells, in which cholesterol reduction after chronic hypoxia decreases Orai1-mediated CRAC currents, suggesting that cholesterol is required for STIM–Orai1 interactions and SOCE. Moreover, using epicholesterol (the enantiomer of cholesterol), the authors concluded on cholesterol specificity and not on lipid raft integrity dependence [208]. Further studies are needed to understand the discrepancies between those studies. Finally, and in a similar mechanism to the one described above for PI(4,5)_2_, STIM1 has a cholesterol-binding domain that is located in the “STIM–Orai activating region” (i.e., the region responsible for interaction with Orai1). However, the interaction between this domain and the inner PM leaflet cholesterol has an inhibitory effect on Orai1 activity, as it reduces the binding between the two proteins [206].

### 3.2. Ca^2+^ Efflux through Exchangers and Selective Pumps

The PM presents two systems for Ca^2+^ extrusion: (i) A low-affinity, high-capacity NCX (Section 3.2.1); and (ii) a high-affinity, low-capacity PMCA (Section 3.2.2). NCX and PMCA vary in relative proportion from one cell type to another, the NCX being particularly abundant in excitable tissues (e.g., heart and brain) where cells experience important Ca^2+^ peaks. Cytosolic Ca^2+^ is also restored to basal levels by sequestration in the ER/SR via the SERCA pump (Section 3.2.3).

#### 3.2.1. Na^+^/Ca^2+^ Exchanger (NCX)

##### Overview and Pathophysiological Implications

The NCX accomplishes Ca^2+^ extrusion by using the electrochemical gradient of Na^+^: During each cycle, three Na^+^ ions flow down their gradient across the membrane in exchange of the counter transport of one Ca^2+^ ion against its gradient. This antiporter is particularly abundant in the PM, the mitochondria, and the ER of excitable cells like in the heart, but is also expressed in other tissues such as the smooth and skeletal muscle, the kidney, and the brain. Structurally, NCX antiporter contains 10 TM domains with a pseudo-symmetry probably resulting from a gene duplication event [209]. Between the pseudo-symmetric halves, a large cytoplasmic loop is inserted containing regulatory domains that have C2 domain-like structures [210]. The binding of Ca^2+^ on those sites generally activates the exchanger, whereas binding of Na^+^ ions has been shown to deactivate it [211].

The NCX does not bind very tightly to Ca^2+^ (i.e., low affinity), but it can transport the ions rapidly (i.e., high capacity), with up to 5000 Ca^2+^ ions per second [212]. Therefore, it requires large concentrations of Ca^2+^ to be effective and is useful for ridding the cell of large amounts of Ca^2+^ in a short time. It plays a major role in neurons after an action potential, in cardiac muscle relaxation, or in excitation-contraction, a.o. However, the result of this channel activation is a brief influx of a net positive charge (Na^+^), thereby causing cellular depolarization. Since the exchanger direction depends on the membrane polarization, it can change its transport direction if the cell is depolarized enough. The ability for the NCX to reverse direction of flow notably manifests itself during the cardiac action potential, as described in [213]. As the transport can occur in both ways depending on the membrane potential, NCX activity has been linked to excitotoxicity, a pathological process in which nerve cells are damaged or killed by excessive stimulation and subsequent abnormal and harmful Ca^2+^ increase [214]. NCX has been shown to play controversial roles, suggesting that it may operate in both forward (therefore, neuroprotective) and reverse (neurodamaging) directions simultaneously in different areas of the cell, depending on the combined effects of the Na^+^ and Ca^2+^ gradients. 

##### Regulation by Lipids

In 1996, Hilgemann et al. showed that PI(4,5)P_2_ activates the NCX1 in cardiac membranes [215]. Four years later, Philipson et al. evidenced that PI(4,5)P_2_ binds to the endogenous XIP region following putative TM helix 5, suggesting that this lipid directly interacts with the transporter as a non-annular lipid [216]. As a consequence, increase of PI(4,5)P_2_ levels can eliminate the inactivation of the exchanger resulting from the decreased [Ca^2+^]_ic_ or increased [Na^+^]_ic_ [217]. On the other side, PI(4,5)P_2_ depletion by PLC-coupled muscarinic M1-receptor agonists strongly inhibits NCX1 current [218]. Additional indirect, and possibly opposing, mechanisms of NCX1 regulation by PI(4,5)P_2_ have also been described. Those include the membrane trafficking of NCX1 and the contribution of the PI(4,5)P_2_–cytoskeleton interactions in its activation [218].

Other acidic PLPs might regulate the activity of NCX1. Indeed, whereas barely any exchange activity is obtained upon reconstitution of NCX1 into pure PC vesicles, a higher activity (about 25% of the value when reconstituted in vesicles from native sarcolemmal membranes) is observed in membranes containing PS, or PA. On the other side, PI and PG induce little activity enhancement in comparison to pure PC vesicles. Addition of cholesterol elevated transport activity to native-like levels for the former two anionic lipids, but not for PI or PG [219]. Those data suggest that PS and PA can interact with NCX1 unlike PG and PI, possibly because the negatively-charged moiety of the former lipids is less buried in the headgroup than the one of the latter [220].

Regarding SLs, Cer has been shown to inhibit NCX influx and efflux modes when expressed in Chinese hamster ovary (CHO) cells. Localized mutations of the exchanger have allowed to evidence that Cer impairs the Ca^2+^-dependent activation of the exchanger while not affecting its Na^+^-dependent activation [221]. Sphingosine seems to ensure a similar regulation as Cer [221]. On the other side, ganglioside GM1 (but not other gangliosides) appears to closely bind to the NCX at the nuclear membrane, activating thereby Ca^2+^ transfer from nucleoplasm to ER [222,223]. The importance of such a regulatory role is demonstrated by the susceptibility to excitotoxicity and apoptosis of neurons in GM2/GD2-synthase KO mice, a phenotype that can be substantially reduced by injection of a semisynthetic analog of GM1 [224].

#### 3.2.2. Plasma Membrane Ca^2+^ ATPase (PMCA)

##### Overview and Pathophysiological Implications

With a complementary activity to the NCX antiporter, PMCA ensures the maintenance of the very low [Ca^2+^]_ic_, thanks to its much higher affinity but much lower capacity. The PMCA pump is a minor component of the total PM proteins (< 0.1%) and is quantitatively overshadowed by the NCX in excitable tissues. However, even in cells in which the NCX predominates, the PMCA pump is likely to be the fine tuner of cytosolic Ca^2+^, as it can work in a concentration range in which the low affinity NCX is inefficient. The PMCA pump is a P-type ATPase, characterized by the temporary conservation of ATP energy in the form of a phosphorylated enzyme intermediate (P-type). There are four PMCA isoforms which exhibit tissue-specific expression [225]. PMCA1 and 4 are expressed in most tissues while PMCA2 and 3 are mostly restricted to the brain, the striated muscle, and the mammary gland. The pump operates with a 1:1 Ca^2+^:ATP stoichiometry. It is predicted to have 10 TM domains, two large intracellular loops, and N- and C-terminal cytoplasmic tails. The catalytic site is sealed by an auto-inhibitory domain, whose binding to calmodulin upon intracellular Ca^2+^ increase will free the site and allow the increase of the PMCA affinity for Ca^2+^ (Kd from ~ 10–20 to < 1 µM) [225]. The pump also presents other modulation sites, like a binding site for 14-3-3 proteins [226], consensus sites for PKA and PKC, and a high-affinity allosteric Ca^2+^ binding site. The PMCA pump is also a substrate of intracellular proteases, as it contains target sequences for calpains [227] and caspases [228].

PMCA isoforms have been shown to be involved into several processes. Notably, PMCA2 is critical for hearing as it dissipates with peculiar kinetics the Ca^2+^ transiently increased by the opening of MSCs. PMCA4 prevents the Ca^2+^ overload and resulting mitochondrial damages in spermatozoids and is crucial to sperm motility and male fertility [229]. This pump isoform is also essential for the modulation of the activity of the neuronal nitric oxide synthase (nNOS) [230] and thus for the control of NO production, itself contributing to the regulation of excitation–contraction coupling of the heart. Consequently, mutations in PMCA have been linked to several oxidative stress-related diseases, like atherosclerosis, diabetes, and neurodegenerative diseases. PMCA defects also have been reported in various cancer types [231].

##### Regulation by Lipids

As for other Ca^2+^ transport proteins, PMCA is modulated by several lipids. First, the acidic lipids PS and PI (but not the neutral PE) increase the activity (V_max_) of the PMCA pump by accelerating dephosphorylation [232] and increasing the Ca^2+^ affinity (i.e., decreased Kd) [233]. The sensitivity of the PMCA to acidic PLPs is located in two domains of the enzyme, one close to the carboxyl terminus and partially shared with the calmodulin-binding site, and the other located between TM domains 2 and 3 [234]. However, the importance of this regulation is debated. On one hand, Carafoli et al. concluded that the erythrocyte pump is probably permanently activated by acidic PLPs to ~ 50% of its maximal activity [235]. On the other hand, Denning and Beckstein consider that, as acidic PLPs have to be at least 40% of the total PLPs to have an effect (a concentration rarely encountered in mammalian membranes), it is quite unlikely that PMCA is regulated by acidic PLPs in vivo [220]. This minimal concentration could, nevertheless, be reached thanks to compartmentation of the pump in acidic PLP-enriched domains, which remain nevertheless to be evidenced.

The PLP derivative DAG is also able to stimulate the PMCA pump, through the induction of an increment in the V_max_ of the enzyme and in the affinity of the protein for Ca^2+^. The activation induced is additive to that produced by PKC, implying that DAG interacts with the PMCA through a different mechanism, which has been shown to be direct [236].

SLs, and especially gangliosides, modulate PMCA activity in a complex pattern. Indeed, in cortical neurons and synaptosomes, polysialogangliosides (e.g., GD1b) stimulate, monosialogangliosides (e.g., GM1) slightly reduce the activity, and non-sialogangliosides (e.g., asialo-GM1) markedly inhibit PMCA2 and 3, predominant in those cells [237,238]. Thus, the sialic acid residues of gangliosides are important for the modulation of PMCA. Besides, the ganglioside oligosaccharide length (e.g., one sialic acid combined with four sugars in GM1 vs. three in GM2 vs. two in GM3) also affects the PMCA activity [238]. Moreover, the effects of gangliosides on the PMCA appear to be isoform-specific. Indeed, in erythrocytes where PMCA1 and 4 are predominant, GD1b, GM1, GM2 at low concentration, and GM3 all stimulate the PMCA activity, while GM2 at high concentration inhibits it [239,240]. The mechanism by which gangliosides modulate the PMCA activity is still discussed. Indeed, while some studies show that the ganglioside effect is additive to that of calmodulin, affecting both the affinity for Ca^2+^ and the V_max_ of the enzyme [238], others suggest that gangliosides interact with the calmodulin-binding domain, activating the phosphatase activity of the PMCA [239]. Moreover, Cer increases both the affinity for Ca^2+^ and the V_max_ of the PMCA [241]. On the other side, the Cer metabolite sphingosine exerts an antagonistic regulation by inhibiting the pump [241].

Finally, as exposed below (see Section 4.3.1), PMCA appears to be associated with cholesterol-enriched domains and the depletion of this lipid abrogates the pump activity in those domains.

#### 3.2.3. Sarco/Endoplasmic Ca^2+^ ATPase (SERCA)

##### Overview and Pathophysiological Implications

Sarco/endoplasmic Ca^2+^ ATPase (SERCA) is a P-type-ATPase that resides at the ER/SR membrane. It transports two Ca^2+^ ions in exchange of the hydrolysis of one ATP molecule and allows to restore luminal ER/SR Ca^2+^ levels [242]. Distinct genes code for the three different pumps, SERCA1–3, generating 10 isoforms by alternative splicing. SERCA1 is mainly expressed in skeletal muscle, SERCA2a in cardiac and skeletal muscles, and SERCA2b and 3 in non-muscle cells [225,243]. The pump contains 10 TM domains and three major domains on the cytoplasmic face: The phosphorylation and nucleotide-binding domains, which together form the catalytic site, and the actuator domain, which is involved in the transmission of major conformational changes. Small molecules, such as phospholamban (PLB; a SR Ca^2+^-ATPase inhibitor, highly expressed in cardiac muscle) and sarcolipin (another small molecular weight protein), modulate SERCA activity according to cellular requirements and/or extracellular signals [225,243]. For example, SERCA is not active when PLB is bound to it. Increased β-adrenergic stimulation induces the phosphorylation of PLB, reducing its association with the SERCA and favoring Ca^2+^ movements. Calreticulin, an ER protein, and calsequestrin, its SR homolog, play also a major role by sequestering Ca^2+^ within the ER/SR and thus reducing the concentration gradient against which the pump has to work. Indeed, the ER/SR exhibits a much higher Ca^2+^ concentration (10,000×) than the cytosol.

The pump process is carried out in two steps featuring two conformations: An E1 phase that allows for the binding of all substrates, and a post-ATP hydrolysis E2 phase that forces the movement of ions through the protein across the membrane. In smooth, skeletal, and cardiac muscles, SERCA exerts two main functions: It decreases Ca^2+^ levels in cytosol to initiate muscle relaxation and it reloads Ca^2+^ into SR, required for muscle contraction. Indeed, during contraction, the sarcoplasm is flooded with Ca^2+^ ions, which indirectly allows for the contraction of muscles through interactions between actin and myosin. It has also been evidenced that the SR regulates membrane excitability by a negative-feedback control step on Ca^2+^ entry. As a consequence of its function, mutations in the pump can result in a life-altering condition in cattle, as pseudomyotonia, a disease that restricts the relaxation of most muscles.

##### Regulation by Lipids

Unlike PMCA and NCX1, the SERCA pump is inhibited by PS and PA when the mole fraction of those acidic PLPs in a PC membrane exceeds 60% [244]. Specifically, maximal levels of ATP binding are reduced below 40% of the native membrane values, without a concomitant reduction in the pump affinity for ATP. Since acidic PLPs do not bind specifically to the protein, one possible explanation is that oligomerization of ATPases renders ATP binding sites inaccessible [244]. Sphingosine also inhibits the SERCA pump in a mechanism that is similar to the thapsigargin one [245]. Finally, the SERCA pump activity also depends on the surrounding membrane cholesterol. Indeed, using microsomes prepared from macrophages, Tabas et al. evidenced that the enrichment in cholesterol of ER membranes (normally cholesterol-poor), i.e. in conditions somehow mimicking atherosclerotic lesions, inhibits SERCA2b [246].

## 4. Hypothetical Mechanisms for the Regulation of Membrane Ca^2+^ Transport Proteins by Lipids 

Several reviews have covered the lipid–protein interaction importance and the ways that those interactions could regulate membrane protein localization and/or activity. We advise the work of Brown [247], Hedger and Sansom [248] or Brügger et al. [32]. The aim of this section is to integrate the major lipid-based regulations of Ca^2+^ transport protein activity. As mentioned in the Introduction, those include: (i) Direct interaction inside the protein with non-annular lipids (Section 4.1); (ii) close interaction with the first shell of annular lipids (Section 4.2); (iii) regulation of biophysical membrane properties directly around the protein through annular lipids (Section 4.3); and (iv) gathering of several proteins into lipid domains (Section 4.4).

Before describing those four mechanisms, we would like to emphasize that the type of protein–lipid interaction will not only depend on the protein properties but also on lipid abundance, structure, membrane localization, and enzymatic modifications, a.o. We chose three examplative lipids, PI(4,5)P_2_, DAG, and cholesterol, to illustrate this statement. First, PI(4,5)P_2_, which is the best-known lipid modulator of Ca^2+^ channels, exchangers, and pumps (as described in Section 3 and in Figure 5) [249], is a minor PLP of the inner PM leaflet and thus its effect on protein regulation is expected to be achieved through direct interactions, as a non-annular or an annular lipid. It is still unclear how membrane proteins bind to this lipid with such a large headgroup extending away from the hydrophobic bilayer into the cytoplasm (up to 17 Å). It was nevertheless suggested that the membrane proteins might approach PI(4,5)P_2_ from the cytoplasmic side, which means that the principal determinants of the binding site would lie in the cytoplasmic loops [250]. The regulation achieved by PI(4,5)P_2_ could also result from the loss of interaction with the protein upon decrease of surrounding PI(4,5)P_2_ levels following PLC stimulation and/or translocation of PI(4,5)P_2_ into and out of specific domains. In support of this hypothesis, it has been suggested that cholesterol levels differently regulate PI(4,5)P_2_ hydrolysis, probably through compartmentation of the latter and related metabolic pathways in rafts [251]. Nevertheless, partitioning of PI(4,5)P_2_ into lipid rafts has been criticized on the basis of energetic considerations. For instance, PI(4,5)P_2_ has a polyunsaturated acyl chain that would not spontaneously partition into ordered rafts [252]. On the other hand, PI(4,5)P_2_ is also involved in the regulation of Ca^2+^ transport through indirect mechanisms, implying the recruitment of regulating/interacting proteins from the cytoplasm in the SOCE mechanism. 

A second example is provided by DAG. This poorly-abundant lipid can potentially act directly on Ca^2+^ transport proteins. However, additional activation and inhibition of desensitization through the activation of PKC, as well as the synthesis of other important signaling molecules (e.g., PA) from DAG and IP_3_, should also be taken into account once integrating the effect of PLC-derived lipid products on Ca^2+^ transport protein activity.

As a third example, one can cite cholesterol, another major Ca^2+^ transport protein regulator. It is expected to easily act as a non-annular lipid in small lipid pockets of proteins, but it is also a key regulator of membrane fluidity and biophysical properties. Besides its modulation as a non-annular or annular lipid, cholesterol could contribute to compartmentalize the different actors of Ca^2+^ signaling in rafts.

### 4.1. Direct Interaction with Non-Annular Lipids

Non-annular lipids are defined based on their interaction with a great specificity with the protein. Most of those interactions, which have been shown by X-ray crystallography, NMR, and MD simulation approaches, have evidenced small cavities that could contain a specific lipid (i.e., lipid pockets), suggesting that once a lipid occupies the pocket, the protein activity is affected. However, resolution of crystal structures of proteins is often not sufficient to perfectly resolve lipid structures, in particular the positions of unsaturated bonds and acyl chain lengths of the hydrophobic moieties [253]. These non-annular lipids generally make contact with the annular lipid molecules and adopt an orientation similar to the one exhibited by the lipids in the bilayer, which could suggest that non-annular lipids incorporate into their binding sites by simple diffusion from the bilayer [254]. However, it is still extremely difficult to experimentally confirm the biological relevance of those pockets. Hence, most of their functions are deduced based on their location in the structure. As a matter of fact, Lee suggested that those lipids could act as traditional cofactors or as “molecular glue”, strengthening the contact between the subunits of oligomeric membrane protein assemblies, a.o. [255].

Multiple non-annular lipid sites have been described in the prokaryotic K^+^ channel KcsA [256] or in G protein-coupled receptors to which cholesterol molecules bind [257]. However, to the best of our knowledge, SERCA pumps are the only Ca^2+^ transport proteins to be shown to bind non-annular lipids so far. The binding of those lipids between the α-helices could affect the rotation/movement that is needed for the protein activity, as suggested by studies of the effects of the drug thapsigargin, a cholesterol-like inhibitor of SERCA (Figure 6) [258]. The drug binds to the inactive E2 intermediate state, in a cleft between TM α-helices M3, M5, and M7, and prevents the conversion to the active E1 state. Indeed, in this state, the space between helices M3 and M7 is much smaller than in the thapsigargin-bound form. Thus, thapsigargin acts as an inhibitor, preventing the relative movement of helices necessary for function [258]. However, and interestingly, even though thapsigargin shares chemical features with cholesterol, Thogersen et al. have shown, by MD simulations, that cholesterol does not bind to the thapsigargin pocket. On the other side, two major cholesterol-binding pockets have been revealed, one of which co-localizes to a position known to bind sarcolipin, a small protein that regulates SERCA [259]. Besides, a pocket for PLPs has been evidenced between the TM helices M2 and M4 on the cytosolic leaflet of the SERCA only in the E2 state [260]. This non-annular site was shown to affect neither the E2 structural integrity nor its stability, and was speculated to become functionally significant during the E2-to-E1 transition of the pump [261]. Therefore, it remains to deepen to which extent non-annular lipid binding to SERCA affects its activity.

### 4.2. Close Interaction with the First Shell of Annular Lipids

The first shell of annular lipids can interact with membrane proteins, mostly thanks to polar interactions between the lipid headgroup and aromatic amino acids [32], which allow those lipids to be co-crystallized with the protein and analyzed by X-ray. Tyrosine residues present in the interfacial region interact with the lipid phosphodiester group either alone (via ion pair or hydrogen bond) or in combination with positively-charged amino acids. Likewise, tryptophan residues are mainly localized in the interfacial region, with the indole group pointing toward the center of the bilayer [262]. The indole nitrogen atom can indeed form a hydrogen bond with the lipid phosphodiester group [263], while a perpendicular orientation of the indole ring can help to stabilize the lipid acyl chains. Besides, the binding can be stabilized by non-polar interactions between the hydrophobic lipid acyl chains and the TM domain [264,265].

Those interactions might have several roles. First, they might orient the protein into the membrane [266]. X-ray data show that the majority of lipids tightly bound to PM proteins are localized on the electronegative side of the membrane. Hence, recognition of protein TM helices by the ER translocon could critically involve direct protein–lipid interactions that help to guide the membrane protein incorporation [267,268,269]. For instance, when a hydrophobic segment emerges from the ribosome, it can intercalate reversibly in two different orientations into the lateral gate. If the hydrophobic sequence is long and the N terminus is not retained in the cytosol by positive charges, or by the folding of the preceding polypeptide segment, it can flip across the channel and subsequently exit it laterally into the lipid phase. If the N terminus is retained in the cytosol and the polypeptide chain is further elongated, the C terminus can translocate across the channel [267].

Second, composition and organization of the lipid headgroup region could affect the structure of a protein penetrating into this bilayer region, because of the requirements of the polypeptide backbone and of any polar residues for hydrogen bonding tending to drive the formation of secondary structures. As differences in the areas occupied by different lipid molecules (e.g., PE occupies a smaller space than PC) are associated with different patterns of hydrogen bonding and hydration in the bilayer, specific lipid composition around the membrane protein might be required. This protein stabilization role explains why less harsh detergents that avoid delipidation are important means to increase the chances of crystallizing membrane proteins [270].

Finally, the close annular lipids might affect the change of protein conformation during activation/auto-inhibition, as this change is often associated with a variation of the amount of associated lipids. This has been shown for the PMCA, with differences depending on the pump isoform. As a matter of fact, activation of PMCA2 and 4 involves a reorganization of the TM region with the removal of lipid molecules from the protein annulus. Although activation of PMCA4 involves the loss of approximately 15 PLP molecules per protein, the loss is of only six PLPs for PMCA2. This fact probably reflects that, during transport of Ca^2+^, PMCA2 changes its conformation to a lesser extent than PMCA4, an isoform that requires a higher calmodulin or PS local concentration for activation [271]. The number of first-layer PLPs also varies during SERCA conformational changes, from 23 to 26 lipid molecules in the cytoplasmic leaflet, as determined by X-ray crystallography [55].

### 4.3. Regulation of Membrane Biophysical Properties Directly around the Protein through Annular Lipids

Besides direct lipid–protein interaction, Ca^2+^ transport proteins can also be influenced by membrane trafficking, interaction with the underlying cytoskeleton/extracellular matrix, and biophysical properties of the lipid environment, such as packing (Section 4.3.1), thickness (Section 4.3.2), or curvature (Section 4.3.3). Although those three membrane biophysical properties are intimately interconnected, they will be described separately for sake of clarity.

#### 4.3.1. Membrane Lipid Packing 

Membrane lipid packing depends on the ratio between small and large polar heads and the ratio between unsaturated and saturated acyl chains. This can be illustrated by several examples. First, the cis-unsaturated oleyl chain (C18:1) occupies a larger volume than the palmitoyl chain (C16:0) because the double bond induces a ‘‘kink’’ in the middle of the chain which lowers the acyl chain packing density [272]. Second, owing to its acyl chain composition, SM forms a taller, narrower cylinder than PC, increasing its membrane packing density. Consequently, at physiological temperature, a SM bilayer exists in a solid gel phase with tightly-packed acyl chains [273,274]. Third, as compared to PG, CL (or bis-PG) has a smaller head-to-tail surface area ratio and has been shown to form CL-enriched domains in bacteria [275,276]. By interfering with acyl chain packing, sterols inhibit the transition of the membrane to the solid gel state, while rigidifying fluid membranes by reducing the flexibility of neighboring unsaturated acyl chains, thereby increasing membrane thickness [277]. Based on membrane packing criteria, the membrane can be viewed as a patchwork with areas characterized by differences in membrane fluidity. The areas of low fluidity are named solid-ordered (So) phases, in which the lipid acyl chains are tightly packed and in which there is a low rate of lateral diffusion. In contrast, the more fluid areas are named the liquid-disordered (Ld) state, which exhibits both low packing and high lateral diffusion. In addition, at the proper concentration, cholesterol may promote liquid-ordered (Lo) phase, which exposes high packing and high lateral diffusion. 

Membrane lipid packing is critical to warrant proper protein sorting. MD simulations of simplified lipid membranes have suggested that prokaryotic membrane proteins form with their adjacent lipids dynamic protein–lipid complexes with up to 50 to 100 lipids that diffuse laterally together [278]. Considering that lipid diffusion rates are significantly reduced within these shells and accepting that membranes are “more mosaic than fluid” [279], it becomes difficult to tell apart an actively recruited annulus from lipids from preexisting Lo domains, in which the lateral mobility of lipids is reduced.

Several Ca^2+^ transport proteins have been shown to be modulated through lipid packing and/or to associate preferentially to Lo or Ld phases. For example, the PMCA pump and calmodulin partition into Lo domains, as observed by density gradient in primary cultured neurons. The Lo-associated PMCA activity is much higher than PMCA activity excluded from these domains. As a consequence, cholesterol depletion abrogates the Lo-associated PMCA activity without any effects on the non-Lo pool [280]. Those results were supported by the observation of Paterson et al. that PMCA has a higher activity when reconstituted with ordered lipids (such as lens fiber lipid or the synthetic 1,2-dipalmitoyl-sn-glycero-3-phosphocholine, DPPC) than into fluid lipids (e.g., 1,2-dioleoyl-sn-glycero-3-phosphocholine, DOPC) [281].

In contrast to PMCA, the SERCA pump is located in the ER/SR membrane, a naturally more fluid environment than the PM due to its low content in complex SLs and cholesterol (3%–6% of total lipids vs. 30%–50% in PM) vs. a high level in unsaturated PCs [273,282,283]. In support of the importance of such a fluid environment for its activity is the observation that reconstitution of purified SERCA in membranes with increased lipid order reduces the rate of Ca^2+^ pumping by decreasing the phosphorylation rate in the catalytic site [284]. Moreover, the cholesterol enrichment of ER membranes by mβCD–sterol complexes results into inhibition of the SERCA2b in macrophages [246]. However, this fluid environment might favor SERCA susceptibility to oxidative damages. Indeed, by increasing the percentage of saturated fatty acids within SERCA’s lipid annulus through diet, LeBlanc et al. showed greater protection to thermal stress (i.e., to oxidative damages) [285].

Besides pumps, several lines of evidence indicate that Ca^2+^ channels are also affected by the surrounding membrane fluidity. As a matter of fact, using mechanical blebbing injury during pipette aspiration-induced membrane stretch (to simulate the damages imposed to the cardiac smooth muscle cells during ischemia, reperfusion, inflammation, a.o.), Joós et al. showed that the resulting lipid membrane disruption and/or fluidization are linked to increased Ca_v_ channel activation, which could explain the Ca_v_ channel leakiness and arrhythmias in pathologies [286]. On the other side, cholesterol increases the L-type Ca_v_ channel currents in arterial smooth muscle cells [287] and cholesterol decrease impairs Ca_v_ channel function in pancreatic β cells [288]. TRP channels (including TRPC1/3/4/5, TRPV1/4, and TRPM8) preferentially segregate into Lo phases at resting state [289,290,291,292]. Interestingly, cholesterol regulates these channel activity in distinct manners. For instance, depletion of cholesterol reduces the activity of TRPV1 [290] and TRPV4 [293], but increases the stimulation of TRPM8 [292]. For further information on TRP channel regulation by biophysical properties, please refer to [118]. Piezo1 channel is also highly dependent on its surrounding lipid packing environment which can be modulated by dietary fatty acids, as recently demonstrated by Vásquez et al. They showed that margaric acid (a saturated fatty acid) inhibits Piezo1 activation by increasing membrane bending stiffness while long chain polyunsaturated fatty acids modulate channel inactivation by decreasing membrane bending stiffness [294].

#### 4.3.2. Membrane Thickness

An important factor that influences the structure and dynamics of membrane proteins is the lateral pressure profile of membranes [32]. It describes the influence of membrane as a solvent of proteins. The highest pressure is at the interfacial region between hydrophobic and hydrophilic areas, because of the high cost of exposing either fatty acyl chains or hydrophobic amino acids to water [255]. This force; therefore, depends on the degree of hydrophobic mismatch at the protein–lipid interface and across the bilayer [295]. A hydrophobic mismatch is observed when the hydrophobic thickness of the lipid bilayer, defined by the distance between opposing headgroups of the inner and outer leaflets (i.e., typically between 35 and 55 Å for the PM, higher than in the ER), does not perfectly match the hydrophobic length of the embedded protein. For an extensive review, please refer to [296].

The requirement of an ideal bilayer thickness for optimal Ca^2+^ transport protein activity has been extensively studied for the SERCA pump. Early studies found a maximal activity for lipids with 18:0 fatty acids (i.e., a bilayer thickness of ~ 30 Å) [297], with a gradual decrease in activity for either thicker or thinner bilayers [298]. Moreover, the effect of bilayer thickness is correlated to the degree of SERCA oligomerization [299]. Those observations are in agreement with the biphasic dependence of SERCA to bilayer thickness, reaching a maximal activity for lipids with 22:1 fatty acids (or ~ 34 Å bilayer thickness) [300]. Those biochemical results were supported by MD simulations, revealing that SERCA can adapt to the thickness of the bilayer through small conformational changes, but that this adaptation is not efficient in thin (~ 22 Å thickness) bilayers made of lipids with 14:1 fatty acids, resulting in defective ATPase function [301]. Moreover, those MD simulations also calculated that, even though the TM domain of SERCA seems to be adequately shielded by a detergent-containing bilayer of ~ 25 Å thickness, the enzyme requires a larger hydrophobic span of ~ 30 Å in the absence of detergent. Contrasting with those data and predictions, Lee suggested that a 30 Å-thick bilayer would locate a Lysine residue (Lys-972) totally within the hydrocarbon core, which seems unlikely. The hydrophobic thickness would have to be ~ 21 Å to locate Lysine-972 at the luminal surface [6]. Supporting this suggestion, LeBlanc et al. showed that SERCA lipid annulus is enriched in short chain fatty acids (12-14 carbons) [285]. Thus, the bilayer thickness appears important for SERCA activity, but contradictory results are obtained concerning the ideal membrane thickness for its V_max_.

Most of the time, the protein tends to localize in a part of the bilayer where the hydrophobic thickness is favorable. If the protein does not find a match, two adaptation mechanisms could occur: (i) Neighboring lipids could adjust to the protein requirements; or (ii) proteins can tilt to hide the hydrophobic part of their TM domain in the hydrophobic part of the bilayer. As a matter of fact, MD simulation studies suggest that there is a mutual adaptation of a membrane protein and the lipid bilayer during conformational changes [301]. The first adaptation mechanism implies that the conformational changes of Ca^2+^ transport proteins during Ca^2+^ exchanges could induce a change in their hydrophobic thickness, which should result in an adaptation of the surrounding lipids. This is needed to decrease the energy barriers to allow the smooth transition between the various steps of the catalytic cycle. This requirement cannot be fulfilled for too thin membranes, whereas for too thick membranes, the hydrophobic mismatch becomes too pronounced for proper interaction between the protein and the lipids. For SERCA, the conformational changes between E1 and E2 states is suggested in some studies to not affect the membrane thickness [6]. However, in X-ray crystallography analysis, the change of conformation has been shown not only to change the thickness of the bilayer (from 30.9 to 33.4 Å) but also to induce a tilt of the entire protein (by approximately 18 °), resulting in a variation of the cross-section of the TM region (from 959 to 1142 Å²) [55].

The second adaptation mechanism is based on the fact that changes in the bilayer thickness, for example, following membrane stretching, could induce modulations of Ca^2+^ transport proteins. Thus, tension is closely linked to bilayer thickness as it can cause changes in it and in lateral pressure profiles, thereby creating hydrophobic mismatch and subsequent adaptive changes in protein conformation that could gate the pore [302]. MSCs are expected to be highly sensitive to the local lateral pressure fields and their modifications were proposed as one of the molecular mechanisms that provides the mechanical force to shift the channels from an open to a closed state [32]. This was shown for the prokaryotic MSCs, where decreased bilayer thickness lowers the channel activation energy [303]. Sens et al. analyzed the gating free energy of those bacterial MSCs and revealed that the quantitative dependence of the gating tension on the length of the lipid acyl tail matches the prediction from elastic bilayer models. Their conclusion not only reveals that the lipid surrounding environment is crucial for the MSC channel activity, but also that the mechanism behind can be understood in terms of a CG elastic model of the lipid bilayer [304]. Besides prokaryotic MSCs, hydrophobic mismatch was also proposed as a gating mechanism of Piezo channels. For those channels though, the mismatch can either originate from membrane thinning following increased tension or from the reduced local curvature around those channels (see Section 4.3.3 for this particular point) (Figure 7) [305]. Mechanical force can be directly transmitted to the channel through lateral tension in the membrane bilayer, whereby the conformation with the greater cross-sectional area is favored under higher tension. This value will be proportional to the work required to open the channel. All those observations indicate that MSCs are gated by stretching the membrane bilayer (i.e., force-from-lipids). However, as discussed in the next section, changes in membrane curvature must also be integrated in the stretching-based gating mechanism of MSCs. Hence, stretching is not the only actor and force conveyed to the channels from the cytoskeleton (i.e., force-from-filament) has to be considered as well [306], but is beyond the scope of this review.

#### 4.3.3. Membrane Curvature

Remodeling of cell shape is accomplished by recruiting specialized proteins which contain motifs able to generate, sense, or stabilize membrane curvature and which act in synergy with changes in the lipid bilayer (see below) and the cytoskeleton. Regarding proteins, three main mechanisms are currently known: (i) Protein crowding and partitioning of TM domains; (ii) direct insertion of small hydrophobic protein motifs between the lipid headgroups (e.g., Bin/Amphiphysin/RVS (BAR) domain containing proteins); and (iii) scaffold by peripheral proteins and their oligomeric assemblies (e.g., clathrin, COPI, or COPII) [307]. Besides proteins, generation of membrane shape by lipids is generally attributed to intrinsic lipid molecular shapes [308] and lipid membrane transversal asymmetry (bilayer couple hypothesis [309]). We will here below focus on those lipid-based mechanisms.

One possible type of molecular curvature sensors are MSCs, whose gating would respond to bending in their local environment. Numerous reports indicate that alterations in local curvature induced by asymmetric incorporation of lipids or conical/amphipathic compounds can change the response of prokaryotic MSCs to pressure [303]. It is important to note that only local curvature seems to have an impact on MSCs. As a matter of fact, using a finite element model, Martinac et al. showed that the global curvature (radius > 0.1 µm) of a patch-clamped membrane on a model MSC has little energetic consequences [310]. However, curvature at the local level (curvature radius < 100 nm) and the direction of bending (i.e., both concave and convex curvatures of the membrane) are able to cause considerable changes in the stress distribution and the pressure profile through the thickness of the membrane. Clearly, the channel sensitivity to the applied force directly dictates the degree with which local curvature impacts on its gating. However, although it is theoretically possible that MSCs sense convex and concave membrane curvature with equal sensitivity, this mechanism would require curvature sensing structures that are symmetrical. This seems unlikely since those proteins do not contain any amino acid sequences with similarity to any known curvature-sensing proteins and Piezo1 shows no symmetrical features with respect to the bilayer plane, as recently revealed by cryo-electron microscopy [307,311]. It is rather proposed that membrane curvature might just be a means to increase the tension sensed by the MSCs [312]. It must be remembered that the contribution of the cytoskeleton was not integrated in those analyses, a factor that contributes to membrane properties [313], and that is predicted to be an important regulator of Piezo1 sensitivity. Indeed, studies reported that inhibition of actin polymerization with cytochalasin D inhibits whole-cell Piezo1 currents evoked by direct stimulus with a glass pipette, but increases opening in cell-attached pressure-evoked currents [314,315]. A particularity of Piezo channels is also that, based on the strongly curved shape of its large blades, it is thought to induce itself a locally-distinct membrane curvature and a, a fortiori, distinct membrane thickness and tension environment [305]. Consequently, Piezo channels might sense changes in the membrane geometry and tension particularly sensitively [316,317,318,319].

### 4.4. Contribution to Ca^2+^ Exchange/Signaling Inside Lipid Domains 

As exposed in the Introduction, lipid domains have been evidenced in a large variety of cells. For a substantial, albeit non-exhaustive, list of examples, see [16] and [320]. Those lipid domains vary in size, abundance, lipid composition, biophysical properties, and regulation by extrinsic factors (including the cytoskeleton) [320]. Due to such diversity, domains could serve as recruitment/exclusion and/or activation/inactivation platforms for specific membrane proteins, thereby participating in the spatiotemporal regulation of dynamic cellular events. We here discuss their potential role in Ca^2+^ signaling by: (i) Gathering several components of the Ca^2+^ signaling machinery; (ii) providing platforms of annular lipids with appropriate biophysical properties around the Ca^2+^ transport protein; and/or (iii) creating a lipid environment that differs in charge through modification of local Ca^2+^ concentration following a primary transient Ca^2+^ influx.

The protein gathering hypothesis is supported by the following lines of evidence. First, analysis of lipid rafts from presynaptic membranes reveals that Ca_v_2.1, but not Ca_v_1.2, co-distributes and interacts with the SNARE complexes present in the domains. Thus, lipid domains might contribute to optimize compartmentalization of exocytosis machinery and the Ca^2+^ influx that triggers synaptic vesicle exocytosis [321]. Second, Ca^2+^ transport proteins have also been spotted at membrane contact sites (MCSs, defined as domains where the ER/SR membrane is tethered to other organelle membranes such as PM, mitochondria, endosomes, a.o.) by specific tether proteins and where lipid transfer proteins and cell signaling proteins are located [249]. For example, in muscles and neurons, the PM Ca_v_ channels communicate with the SR-associated RyR channels at MCSs [322]. At the apical pole of polarized epithelial cells, all Ca^2+^ signaling proteins are clustered in MCSs where they form complexes [323,324]. Thus, physiological cell stimulation evokes Ca^2+^ signals confined to the apical pole [325]. Although such regulation by lipids suggests the localization of ion channels and transporters at MCSs, this remains to be clearly demonstrated.

The lipid platform hypothesis is based on the specific biophysical properties of lipid domains, which might in turn contribute to regulate Ca^2+^ transport protein activity. One class of Ca^2+^ transport proteins that appears to depend on lipid domain biophysical properties are the TRP channels, as supported by the following lines of evidences. First, upon prevention of the association of TRPM8 with lipid rafts, menthol and cold-mediated responses are potentiated, suggesting that this association is a key regulator for TRPM8 activity. Similarly, lipid raft disruption shifts the threshold for TRPM8 activation to higher temperatures [292]. Cholesterol suppresses TRPM3 constitutive and stimulated activity [326], suggesting its role as a negative modulator of this channel, even if a recent study reported opposite results [127]. Likewise, the effect of TRPV1 segregation in lipid rafts is still unclear. Indeed, it seems to regulate the protein level at the cell surface without any effect on the channel response [290,327,328]. However, another group has, rather, suggested that direct interaction with cholesterol (i.e., that might not require segregation in rafts) inhibits TRPV1 opening [329]. Thus, all the above studies seem to indicate the importance of lipid rafts and/or cholesterol in TRP channel activity, but further studies are needed to clarify some discrepancies. The activity of MSCs also appears to be regulated by lipid rafts. In 2015, Hu et al. showed that force transfer, and thus sensibility of MSCs, including Piezo channels, is regulated through stomatin-like protein-3 (STOML3) and its binding to cholesterol [330]. Since STOML3 is detected in lipid rafts, this could suggest that Piezo channels are also associated with rafts. Finally, lipid domains could also modulate the activity of Ca_v_ channels. For instance, disruption of caveolae by cholesterol depletion perturbs the response of Ca_v_1.2 activity in heart cells and the response of Ca_v_2.1 to β2-adrenergic stimulation [331,332]. Moreover, Ca_v_1.2 channels have been associated with caveolin-3 (a major caveolae protein) and signaling molecules from the β2-adrenergic pathway [331] (Figure 8A).

The Ca^2+^-induced domain formation hypothesis is based on the observation that localized membrane domains that differ in charge can be created in cells through the modification of local Ca^2+^ concentration by transient Ca^2+^ influx from membrane channels like TRPs. Those Ca^2+^ ions in turn interact with negatively-charged membrane lipids, forming Ca^2+^-induced domains [334]. For example, TRMP7 drives the formation of Ca^2+^ domains during invadosome formation in neuroblastoma cells [335] and upon migration of human embryonic lung fibroblasts [333]. In the latter study, the Ca^2+^ domains are most active at the leading lamella of migrating cells (Figure 8B) [333]. It remains to determine if such localized membrane charge in the inner PM can in turn affect the organization of the lipids in the outer PM leaflet and recruit and/or activate other membrane Ca^2+^ transport proteins, thereby contributing to Ca^2+^ exchanges.

## 5. Lipid and Ca^2+^ Transport Alterations in Physiological Aging and Diseases

Like during physiological aging, several diseases have been linked to, or are exacerbated by, altered Ca^2+^ signaling. One can cite neurodegenerative diseases (e.g., Alzheimer and Huntington diseases), cardiac or muscular diseases (e.g., Duchenne muscular dystrophy), and RBC deformation-linked diseases (e.g., hereditary spherocytosis and elliptocytosis), a.o. Those diseases have also been linked to altered membrane lipid composition, organization in lipid domains, and/or biophysical properties, which might be the missing link between the initial perturbation and the Ca^2+^ exchange perturbation.

For example, in Huntington disease, the mutation in the huntingtin protein is believed to alter lipid metabolism, and perturbation of PM fluidity represents an early event of the disease onset [336]. On the other side, the disease is also linked to a Ca^2+^ dyshomeostasis, which is believed to be the main cause of the disease phenotype [337].

Another example relates to RBC membrane fragility diseases. We have recently described a close link between Ca^2+^ exchanges during erythrocyte deformation and the modulation of the abundance of several lipid domains at their surface [338]. Moreover, in RBCs from patients with spherocytosis and elliptocytosis resulting from cytoskeleton defects, we evidenced that Ca^2+^ intracellular content and exchanges as well as PM lipid organization in domains are altered. In elliptocytosis in particular, the lipid domains involved in Ca^2+^ exchange exhibit altered cytoskeleton anchorage and cholesterol enrichment, leading to Ca^2+^ exchange impairment upon deformation and increased RBC fragility. Those effects are closely linked to the alteration of the RBC membrane lipid composition by oxidative stress and upregulated plasmatic acid sphingomyelinase (our unpublished data).

Cell membrane fragility, mechanical alteration of cytoskeletal-extracellular connection, modification of Ca^2+^ homeostasis and oxidative stress are similarly observed in Duchenne muscular dystrophy [339]. This progressive neuromuscular disease is characterized by muscle degeneration and is caused by the deficiency of dystrophin, a huge protein that constitutes the link between the cytoskeleton and the extracellular matrix. As the disease progresses, fibrosis and atrophy become apparent and regenerating fibers are less frequent. Importantly, many proteins involved in the pathophysiological production of Ca^2+^ and reactive oxygen species (ROS) in dystrophic muscle have been shown to localize in caveolae. Those include TRPC1/4, NADPH oxidase 2, and the small GTPase Rac1. Those observations suggest that caveolae could act as signaling platform for Ca^2+^ and ROS production. For an extensive review on early damage pathways in muscular dystrophy, see [339].

The PMCA pump appears to be altered in neurons during aging [340] and in neurodegenerative diseases like Alzheimer’s disease [341]. For the latter, the inhibition was shown to be induced by the Tau protein, whose effect surprisingly decreases with aging. Interestingly, age-dependent changes in ganglioside total content and species were observed in rat brain synaptic PMs, with an important increase of the monosialoganglioside GM1, without any modification of the fatty acid composition [237]. These findings are consistent with the age-associated, high-density clustering of GM1 at presynaptic terminals reported previously [342]. As gangliosides are major regulators of PMCA (see Section 3.2.2), the authors suggested that an alteration of its activity might result from ganglioside perturbation during aging. Hence, the inhibition due to ganglioside perturbation might take over the alteration caused by the Tau protein in neurodegenerative diseases. Finally, alterations caused by perturbation of membrane lipid composition could also be linked to increased sensitivity of the PMCA to oxidative stress upon aging [343].

## 6. Lipids as Therapeutic Targets in Ca^2+^ Transport-Related Diseases

Induction of PM lipid alterations might help to treat diseases linked to Ca^2+^ transport alterations. As a first example, one can cite the thermosensitive TRP channels expressed in nociceptive sensory neurons (TRPV1–4, TRPM2/3/8, and TRPA1) and which contribute to inflammatory and neuropathic pain conditions. The current first-line therapeutics for pain provide only partial relief and have harmful side effects when used long-term. Consequently, identification of new molecular targets for analgesic drug development is crucial and urgent. In this context, lipid mediators represent attractive targets. Among them, PIP_2_ is a choice modulator, although nowadays limited to experimental systems. Considering that major TRP channels specifically partition in rafts [344], disruption of the later by ω-3 fatty acids could modulate the signaling events and exert anti-inflammatory actions. This supports the use of fatty acids as nutraceuticals for the treatment of inflammatory disorders such as rheumatoid arthritis, inflammatory bowel disease, and asthma [345,346]. In a disease linked to Piezo1 gain-of-function mutations (i.e., hereditary stomatocytosis) dietary fatty-acid supplementation has been shown to abrogate the phenotype in mice and could be proposed as a treatment [294].

## 7. Conclusions and Challenges for the Future

In this review, we have highlighted that, through protein interaction with the annular shell of lipids or with immersed lipids in cavities and clefts at their surface, the lipid environment contributes to modulate the activity of Ca^2+^ channels, exchangers, and pumps. Among those lipids, cholesterol, acidic PLPs, SLs, and metabolites appear to play key roles. Elucidation of such lipid–protein interactions was possible thanks to X-ray diffraction, electron crystallography, and/or NMR, which allow to determine the atomic level structure of lipid-binding sites on membrane proteins. Thanks to those techniques, over 100 structures of membrane proteins containing electron density interpreted as bound lipid molecules were already available in 2014 [53], and we can expect an exponential increase of this amount from that time. MD simulation is a powerful way to extend those analyses to lipid-binding sites of less affinity and to simulate the repercussion of membrane composition/property modifications on the protein structure and functionality [248]. The major challenge of this technique is to transcribe in the model the complexity of the biological membrane, with its extremely diverse composition and asymmetry. However, over time, more and more complex membranes are modeled, nowadays including over 60 different lipids and an asymmetric distribution between leaflets [347], and we can hope that this will soon become the norm. We believe that the combination of imaging studies, protein structural biology, and MD simulations will contribute to help the scientific community to elucidate the mechanistic behind protein–lipid interactions upon Ca^2+^ exchanges and their pathophysiological relevance.

Adding to the above complexity, the link between lipid and proteins can also be seen in terms of (modification of) membrane biophysical properties, lipids creating an appropriate biophysical environment in terms of membrane fluidity, thickness, or curvature. Moreover, lipid domains could provide platforms for communication between membrane proteins and their downstream targets and/or modulators. However, the latter mechanism is nowadays still hypothetical, probably resulting from the difficulty to evidence lipid domains in living cells due to a long-term limitation of probes and imaging techniques [16]. Today, thanks to the analysis of a same-target lipid by several complementary probes, for their respective advantages and drawbacks (when available), and thanks to the development of new technological approaches (e.g., super-resolution techniques), lipid domains have been reported in a variety of cells (reviewed in [16,320]). One major issue is whether lipid domains can be generalized or if they are restricted to cells exhibiting particular membrane lipid and protein composition, biophysical properties, and/or membrane–cytoskeleton anchorage. In addition, determination of how membrane biophysical properties and extrinsic factors could confine proteins into domains is needed.

## Figures and Tables

**Figure 1 biomolecules-09-00513-f001:**
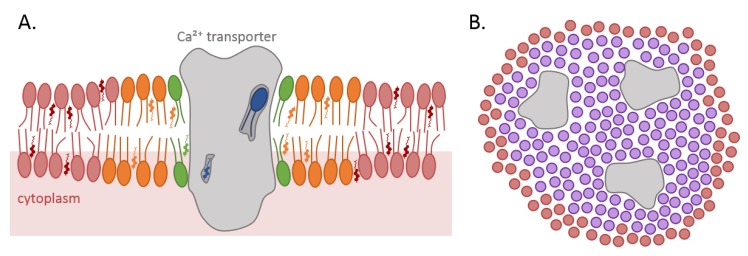
Simplified representation of potential modes of interactions between lipids and Ca^2+^ transport proteins. (**A**) Side view of a single protein anchored in the membrane. First, non-annular lipids (blue) interact directly with the protein (grey) inside its structure. Second, annular lipids from the first shell (green) can closely interact with the protein. Third, the annulus of surrounding lipids (orange) can also modulate protein activity through its biophysical properties. (**B**) Top view of several proteins gathered in one lipid domain (purple), as a fourth potential mode of interaction. Bulk lipids are represented in red.

**Figure 2 biomolecules-09-00513-f002:**
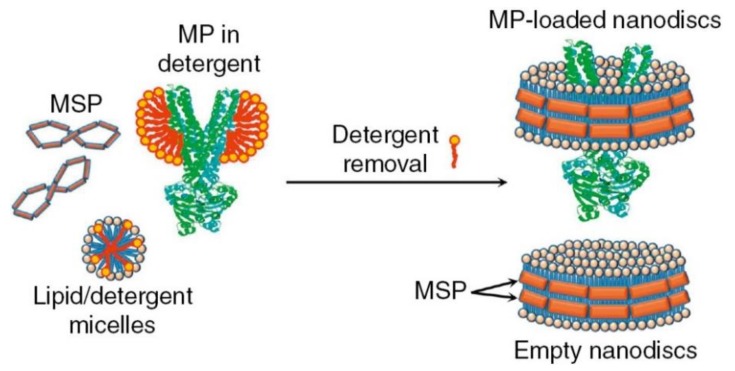
Traditional assembly of a nanodisc with a membrane protein (MP). Reproduced from [43]. MSP, membrane scaffold protein.

**Figure 3 biomolecules-09-00513-f003:**
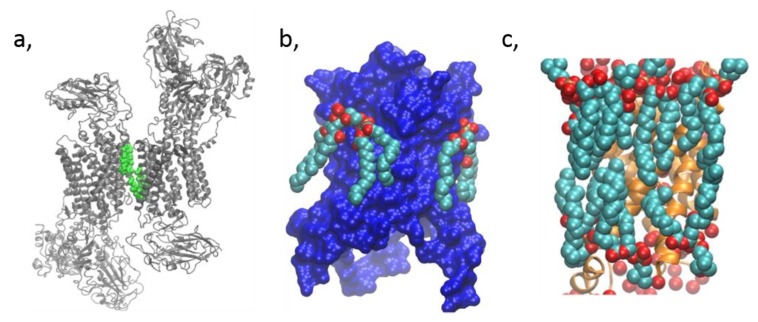
X-ray crystal structures of: (**a**) The porcine Na^+^/K^+^ ATPase (grey) with bound cholesterol molecules (green), most certainly as non-annular lipids; (**b**) the bovine ADP/ATP carrier (blue) with annular-bound diphosphatidylglycerol molecules (cyan); and (**c**) the lens aquaporin (orange) with several phosphatidylcholine (PC) molecules (in cyan) in a configuration approximating the bilayer. Those images were reproduced from [53].

**Figure 4 biomolecules-09-00513-f004:**
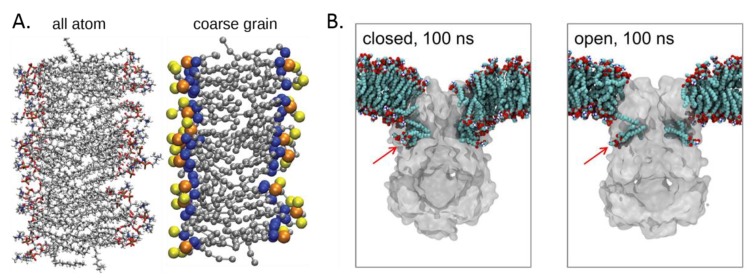
Molecular dynamic (MD) simulations. (**A**) A 1-palmitoyl-2-oleoyl-sn-glycero-3-PC (POPC) bilayer represented in an all-atom model (**left**) vs. a Coarse-grained (CG) model (**right**). Reproduced from [76]. (**B**) Prokaryotic mechanosensitive channel of small conductance (MSCS)–lipid interactions upon closed (**left**) and open (**right**) conformations in CG MD simulations. The red arrow shows the lipid-binding cavity whose occupancy changes between the closed and the open conformation. Original MD simulation from [77].

**Figure 5 biomolecules-09-00513-f005:**
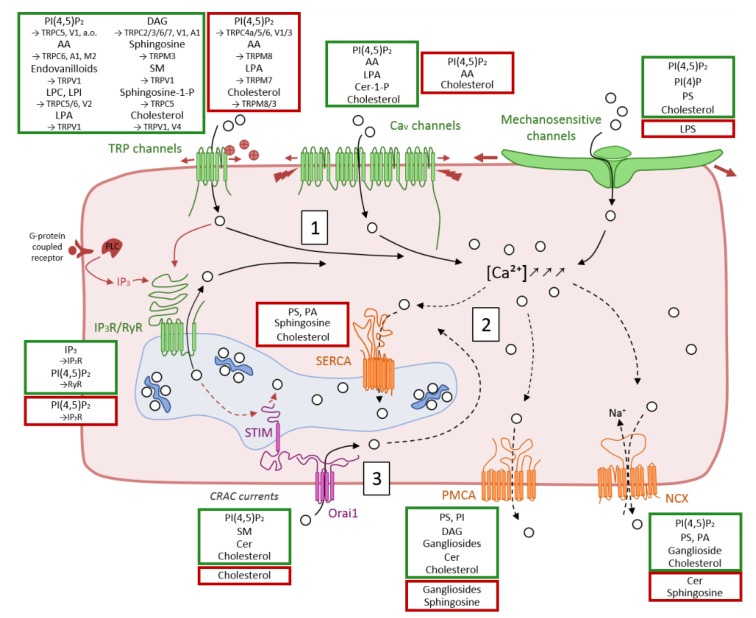
Main Ca^2+^ transport proteins and their known regulation by lipids. (**1**) Following a diversity of stimuli (in brick red), Ca^2+^ (white dots) influx into the cytosol will occur through channels at the plasma membrane (PM) or at the endo-/sarcoplasmic reticulum (ER/SR) membrane (in green). (**2**) Ca^2+^ increase will activate transport proteins (in orange) that export Ca^2+^ out of the cell or inside the ER/SR (in light blue), where it is sequestrated by calreticulin/calsequestrin (in blue). (**3**) Store-Operated Ca^2+^ Release-Activated Ca^2+^ (CRAC) currents (in pink) occur through the PM upon depletion of the ER/SR Ca^2+^ stores. Continuous and discontinuous arrows point to cytosolic Ca^2+^ increase and decrease, respectively. All Ca^2+^ transport proteins are represented based on their known structure (see description in the sections below), except for the Piezo mechanosensitive channel, for which a simplified view of the trimer is shown. For inositol-3-phosphate receptor (IP_3_R) and ryanodine-receptor (RyR) tetramers and Orai1 hexamer, only one monomer is represented, while for Ca_v_ channels only the α-subunit is drawn. Regarding regulation of those Ca^2+^ transport proteins, only lipids are depicted: Those stimulating the transport proteins are boxed in green and those inhibiting them are in red.

**Figure 6 biomolecules-09-00513-f006:**
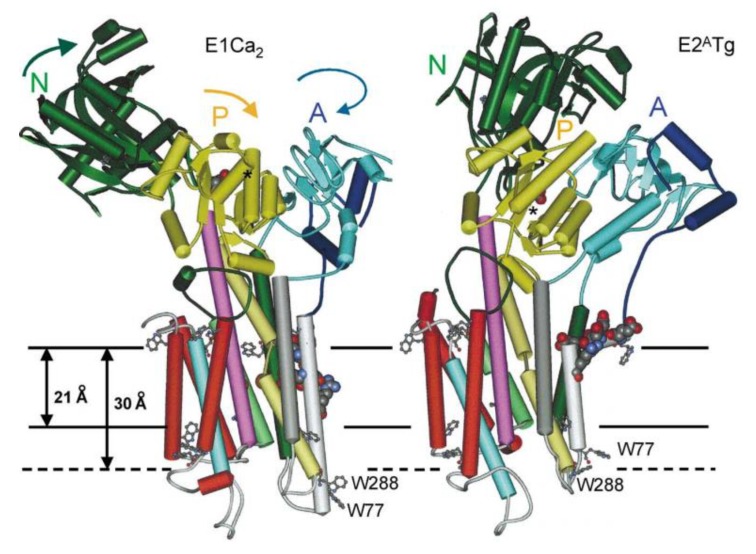
Comparison of the structures of the Sarco/Endoplasmic Ca^2+^ ATPase (SERCA) in its Ca^2+^-bound (E1Ca_2_) and Ca^2+^-free, thapsigargin-bound (E2^A^Tg) forms. Thapsigargin binds in a pocket between the helices M3 (dark grey), M5 (lilac), and M7 (red); therefore, preventing the conformational change to the active E1 state, in which the pocket is smaller. Reproduced from [258].

**Figure 7 biomolecules-09-00513-f007:**
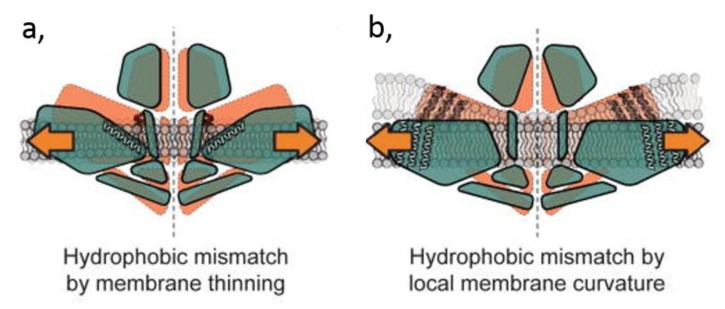
Two of the proposed gating mechanisms for Piezo channels. Alterations of the membrane thickness originating from membrane thinning (**a**) or reduced curvature through stretching (**b**) result in hydrophobic mismatch and adaptive changes of Piezo conformation. Reproduced from [305].

**Figure 8 biomolecules-09-00513-f008:**
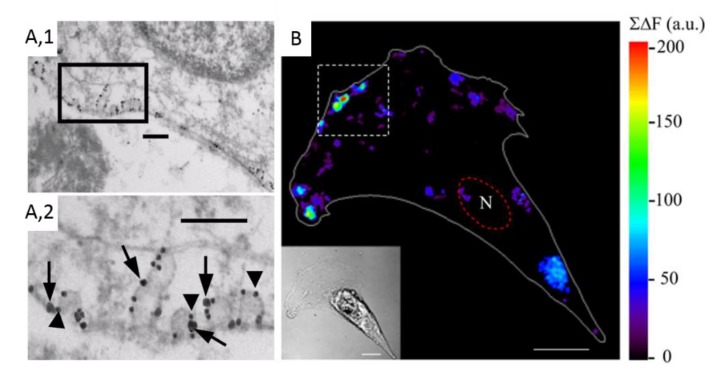
Illustration of the interplay between lipid domains and Ca^2+^ signaling. (**A**) Association of a Ca^2+^ channel with caveolae (A,1; zoom in A,2). Immunogold colocalization of the Ca_v_1.2 subunit of L-type Ca^2+^ channel (large particle, arrows) and caveolin 3 (small particle, arrowheads) in the caveolae in isolated mouse cardiomyocytes. Scale bars, 200 nm. Reproduced from [331]. Copyright (2006) National Academy of Sciences, USA. (**B**) Ca^2+^ domains in migrating cells. Local Ca^2+^ increases (ΣΔF) were summed over 30 consecutive images acquired at 6 s intervals. N points to the nucleus of the fibroblast. Scale bar, 15 µm. Reproduced from [333].

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
