# Peer review of "Regulation of Membrane Calcium Transport Proteins by the Surrounding Lipid Environment"

_biomolecules, 2019, doi:10.3390/biom9100513_

Round 1

Reviewer 1 Report

The manuscript by Conrad and Tyteca represent a high quality, comprehensive review of protein-lipid interactions which could play a role in regulation of the calcium ion transporters. Enough evidence for the phenomenon itself, about various methodologies used for its detection and possible molecular mechanisms of its function, as well as its potential physiological relevance, are provided to the reader. The overall readability of the manuscript could be further improved by several changes suggested below.

Major points

The Chapter order should be re-considered. References to parts of Chapter 5 occur not less than eleven times in various parts of the text prior to Chapter 5 begins, mainly in Chapter 4. It makes the Chapter 5 the “impatiently awaited finale” of the manuscript, but also shifts Chapters 2-4 into the position of just “enormously extended introduction”. In contrast, there is none recall to Chapter 4 in Chapter 5. In addition, the main mechanisms for the regulation of Ca2+ transporters by lipids, i.e. the subject matter of Chapter 5, are repeatedly enumerated earlier in the text (Abstract, Chapter 1, Chapter 4). The idea of the lipid raft – DRM equivalence, as well as that of bipolar partitioning of biological membranes into “raft” or “non-raft” lateral domains, is obsolete. The authors should adequately reformulate the statement in rows 374-5 and others later in the text. Additionally, the raft/non-raft view of the plasma membrane collides with their presentation of the membrane as a “patchwork with areas characterized by differences in membrane fluidity” (rows 1054-5), which includes also solid-ordered (ordered, sphingolipid-enriched, but apparently “non-raft”) microdomains into the consideration, and with the part 5.4, mentioning even broader variety of membrane microdomains generated by other than “raft” mechanisms. It is also unclear why “…cholesterol could contribute to compartmentalize the different actors of Ca2+ signaling in lipid rafts“ (rows 819-20, part 4.2), but similar statement about sphingolipids is missing in part 4.3. Nowadays, lipid raft theory is understood as one of the possible mechanisms of membrane microdomain formation, similarly to for example Kusumi’s membrane-cytoskeleton fence and picket model. Why the former mechanism is mentioned so frequently in the text, while the latter is omitted at all, although its relevance for Ca2+ is evident, for example in relation to cytoskeleton defect-coupled RBC membrane fragility diseases, discussed in Chapter 6? This discrepancy should be balanced by the authors.

Minor points

Row 36: Why plasma membranes are mentioned exclusively? Shouldn’t rather be “different membranes of the same type/kind” or something like that? Row 43, 44: A subject mismatch? “…we focus on calcium (Ca2+) ions, as major MESSENGERS in the cell signaling. Those SIGNALS are generated…“ Which signals? Complete the legend for Figure 1: (i) What is meant by the „red brick“? (ii) What represent the blue structures inside the ER (is the light blue thing ER, right?) accumulating Ca2+ ions? (iii)The yellow thing with the DNA symbol is the cell nucleus? Why it is included? Concerning the NCX Na+/Ca2+ exchanger, there is written in rows 290-1 the following: “…the result of this channel activation is a brief influx of a net positive charge (K+), thereby causing cellular depolarization“. Shouldn’t be replaced K+ for Na+? In part 3.1.2 (rows 391-406), references should be added. Similar to the other techniques mentioned, a brief explanation of the Airyscan imaging principle should be provided (rows. 407-15). The introducing paragraph of Chapter 4 (rows 638-46) could be improved: The statement in the first sentence is valid for “membranes of eukaryotic cells” only. In the list of acidic phospholipids, the cardiolipin is missing. Do authors mean just plasma membranes here, or membranes excluding inner mitochondrial membrane, or…? Similarly in the sentence about sphingolipids, only sphingomyelin and glycosphingolipids are mentioned. Does the paragraph describe mammalian membranes exclusively? This should be clearly specified, as bacterial and yeast examples are given elsewhere in the text, too. Rows 1045-8: Does another example exist, in which the area-per-lipid could be compared between the two phospholipids with acyl chains of the same length? Comparison of C18:1 with C16:0 mixes the two parameters together. Statement in rows 1058-1060 should be rephrased. In general, cholesterol-induced liquid ordered phase can exist alone, i.e. without the lateral separation from the liquid disordered phase. See for example Ipsen et al. 1987, Phase equilibria in the phosphatidylcholine-cholesterol system, Biophys. Acta 905:862–72. Row 1238: definition of DRM abbreviation should be moved to the row 375, where it is missing. Consider explaining the terms ”annular lipids” / “non-annular lipids” earlier in the text. The terms are used through the whole manuscript incl. Abstract. For significant part of potential readers could be Chapter 4 (rows 663, 672) too late to explain them. Correct typos: row 81 – “stroke”; row 547 – “biased”; row 803 – “its modulator”

Author Response

Dear Editor,

Let me first to thank you very much for your quick and constructive feed-back on our manuscript (ID: biomolecules-580732).

We are grateful for the in-depth analysis of the two Reviewers and of the academic Editor. All comments were taken into serious consideration, as detailed hereafter.

We sincerely hope you will agree that all criticisms and suggestions have been satisfactorily addressed.

I thank you for your kind attention,

Sincerely yours,

Donatienne Tyteca

-----------------------------------------------------------------------------

Reviewer 1

We are grateful to this Reviewer for his/her in-depth analysis and clear criticisms. All were taken into serious consideration, leading to further improvement of the manuscript.

We specifically:

Reconsidered the Chapter order by moving key concepts of previous Chapter 5 (mechanisms) in the Introduction section while merging previous Chapters 2 (overview of calcium transport proteins) and 4 (lipid regulation) together and placing previous Chapter 3 (methods) at the beginning of the manuscript. Revised the description regarding lipid rafts while integrating the regulation of lipid domains by membrane and cytoskeletal proteins besides lipid-based mechanisms and the Kusumi’s based model. Those statements are now provided in the Introduction section (because they are needed all along the manuscript). Integrated all the minor points highlighted by the Reviewer. They appear in green in the revised version of the manuscript.

Reviewer 2 Report

File attached.

Author Response

Dear Editor,

Let me first to thank you very much for your quick and constructive feed-back on our manuscript (ID: biomolecules-580732).

We are grateful for the in-depth analysis of the two Reviewers and of the academic Editor. All comments were taken into serious consideration, as detailed hereafter.

We sincerely hope you will agree that all criticisms and suggestions have been satisfactorily addressed.

I thank you for your kind attention,

Sincerely yours,

Donatienne Tyteca

----------------------------------------------------------------------------

Reviewer 2

We are grateful to this Reviewer for his/her in-depth analysis and clear criticisms. All were taken into serious consideration, leading to further improvement of the manuscript.

We specifically:

Reconsidered the Chapter order by moving key concepts of previous Chapter 5 (mechanisms) in Chapter 1 (introduction) while merging previous Chapters 2 (overview of calcium transport proteins) and 4 (lipid regulation) together and placing previous Chapter 3 (methods) at the beginning of the manuscript. Hence, once possible, we decreased the information provided in the previous Chapters 2-4. Revised the title, which is now: ‘Regulation of membrane calcium transport proteins by the surrounding lipid environment’. Revised grammar, phrasing and/or spelling throughout the review. Extended the description of the molecular simulation methods to the work of Muller et al, 2019; Lyman et al, 2018 and Fosso-Tande et al, 2017.